

# Rheological stratification in impure rocksalt during long-term creep: morphology, microstructure and numerical models of multilayer folds in the Ocnele Mari salt mine, Romania

Marta Adamuszek[1], Dan M. Tămaș[2], Jessica Barabasch[3], Janos L. Urai[3]

[1]Computational Geology Laboratory, Polish Geological Institute - NRI, Warszawa, Poland
[2]Department of Geology and Research Center for Integrated Geological Studies, Babeș-Bolyai University, Romania
[3]Institute for Tectonics and Geodynamics, RWTH Aachen University, Germany

*Correspondence to*: Marta Adamuszek (marta.adamuszek@pgi.gov.pl)

## Abstract

Analysis and prediction of deformations in salt tectonics and salt engineering require information about the mechanical properties of rocksalt at time scales far longer than possible in the laboratory. It is known that at laboratory time scales, rocksalt samples with different composition and microstructure show a variance in steady state creep rates, but it is not known how this
variance is manifested at low strain rates and corresponding deviatoric stresses. Here, we aim to quantify this from the analysis of multilayer folds that developed over geological time scale.

We studied excellent exposures of layered, folded rocksalt in the Ocnele Mari salt mine in Romania. The formation is composed of over 90% of halite, while distinct multiscale layering is caused by variation in the fraction of impurities. Regional tectonics and mine-scale fold structure are consistent with deformation in a shear zone, after strong shearing in a regional
detachment, forming over ten meter-scale chevron folds of a tectonically sheared sedimentary layering, with smaller folds developing on different scales in the hinges. Morphology of the fold pattern at various scales clearly indicates that during folding the sequence was mechanically stratified. The dark layers contain more impurities and are characterized with a more regular layer thickness as compared to the bright layers and, thus, are inferred to have higher viscosities.

Optical microscopy of Gamma-decorated samples shows a strong shape preferred orientation of halite grains parallel to the
foliation, which is reoriented parallel to the axial plane of the folds studied. Microstructures indicate dislocation creep, together with extensive fluid-assisted recrystallization and strong evidence for solution-precipitation creep indicative for linear (Newtonian) viscous rheology during folding. Deviatoric stress during folding was lower than during shearing in the detachment, around 1 MPa.

We investigate fold development on various scales in a representative multilayer package using finite element numerical
models, constrain the relative layer thicknesses in a selected outcrop and design a numerical model. We explore the effect of





different Newtonian viscosity ratios between the layers on the evolving folds on different scales. Through the comparison of the field data and numerical results, we estimate that the effective viscosity ratio between the layers was larger than 10 and up to 20. Additionally, we demonstrate that the considerable variation of the layer thicknesses is not a crucial factor to develop folds on different scales. Instead, unequal distribution of the thin layers, which organize themselves into effectively single

layers with variable thickness can trigger deformation at various scales.

Our results show that impurities can significantly change the viscosity of rocksalt deforming at low deviatoric stress and introduce anisotropic viscosity, even in relatively pure, layered rock.

## 1    Introducton

Understanding the rheology of rocksalt during long-term deformation is of great significance in modelling salt tectonics and
in salt engineering, e.g., salt diapir evolution, evolution of salt basins, designing, operation, and abandonment of underground storage caverns and nuclear waste repositories. Quantifying salt tectonic flow requires extrapolation of experimentally derived flow laws to strain rates much lower than those attainable in the laboratory (Herchen et al., 2018). This extrapolation must be based on an understanding of the microscale deformation mechanisms operating under these conditions, and on integrated studies of natural structures with experimental work (Urai et al., 1987; Weinberger et al., 2006). Reviews are provided by
Carter and Hansen (1983) and Urai et al. (2008a).

Under geological conditions, rocksalt deforms by a combination of dislocation creep and solution-precipitation creep processes, which are described by constitutive equations relating strain rate to stress, grain size and temperature. In-situ differential stress in the deep subsurface, using subgrain size piezometry, indicate values usually in the range of 0.5 to 2.3 and sometimes as high as 5 MPa MPa, (Schléder and Urai, 2005, 2007, Rowan et al., 2019) so that in-situ stress in rocksalt is
nearly isotropic. Under these conditions, dynamic recrystallization maintains a stress-dependent steady state grain size and power-law rheology is common with an n-value of about 4.5. Microstructural studies of subgrains and recrystallised grains in naturally deformed rocksalt also show, in agreement with experiments, that during fluid-assisted dynamic recrystallization of salt in nature (water content >10 ppm), the grain size can adjust itself so that the material deforms close to the boundary between the dislocation and pressure solution creep fields (Ter Heege et al., 2005). In the dislocation creep regime, a large
number of laboratory experiments have shown that depending on microfabric and impurities, rocksalt power law creep rates can vary by several orders of magnitude (the so called "Kriechklassen") (Hunsche et al., 2003).

However, at low deviatoric stresses and at long time scales, when the grain size is lower than the steady-state grain size for the current deviatoric stress, pressure solution is the dominant deformation mechanism, and rheology is Newtonian viscous (Urai et al., 1986; Bérest et al., 2019). In recent years, advances in understanding of the deformation mechanisms and microstructural
processes have been reported, based on developments in microstructural and textural/orientation analysis using electron backscatter diffraction (EBSD), microstructure decoration by Gamma-irradiation, Cryo-SEM and other methods. Samples from a wide range of subsurface and surface locations have been studied (e.g., Schléder and Urai, 2005, 2007; Schoenherr et



al., 2007; Urai and Spiers, 2007; Urai et al., 2008b; Leitner et al., 2011; Závada et al., 2012; Kneuker et al., 2014; Thiemeyer, 2015; Thiemeyer et al., 2016).

Various deformation structures or processes in nature can be used to infer long-term rheological properties of rocks (e.g., (Price and Cosgrove, 1990; Talbot, 1999; Kenis et al., 2005). A number of methods have been used to constrain long-term rheology of rocksalt: 1) surface displacement field in areas of active salt tectonics and, in the areas, where removal of ice sheets has led to a change of overburden load (e.g., Weinberger et al., 2006; Mukherjee et al., 2010), 2) sinking or ascent of dense rocks in the rocksalt (e.g., Weinberg, 1993; Burchardt, 2012; Burchardt et al., 2012; Li et al., 2012; Adamuszek and Dabrowski, 2019),

3) cavern convergence (e.g., Bérest et al., 2017; Cornet et al., 2018), 4) naturally flowing salt at surface (e.g., Talbot, 1979; Talbot et al., 2000), 5) development of finger-like structures (e.g., Słotwiński et al., 2020). The analyses usually assume either linear viscous rheology or power-law creep. However, the problem of which of these rheologies best describes long-term behaviour of the rocksalt has not been satisfactorily solved.

In tectonic models, mechanical behaviour of evaporite successions is usually assumed homogeneous and isotropic and

mechanical properties of the succession are approximated with the rheology of rocksalt. However, evaporites are often layered with common intercalation of rocks such as bittern salt (carnallite, bischofite, epsomite) or anhydrite. Bittern salt are much weaker than rocksalt (Urai, 1983, 1985; Urai and Boland, 1985; Urai et al., 1986; Urai, 1987b, a; Schenk and Urai, 2005; Słotwiński et al., 2020). A useful rule of thumb is that bittern salts have a 100 to 1000 times lower effective viscosity than rocksalt and flow 100 to 1000 times faster than rocksalt under a given differential stress, forming effective bedding parallel

detachments (Rowan et al., 2019). Rheology of anhydrite in the subsurface is not well constrained by extrapolation of laboratory experiments, probably because in nature solution-deposition processes are much more important. Constraints are provided by folds of anhydrite in rocksalt, which are invariably concentric and point to an effective viscosity about 10-100 times that of rocksalt (Schmalholz and Urai, 2014; Adamuszek et al., 2015). In contrast, in layer-parallel extension anhydrite boudinages and readily fails in extension, possibly because of the high pore pressures sustained by the surrounding

impermeable rocksalt. In summary, in layered evaporites, rheological contrast can be as high as five orders of magnitude (Rowan et al., 2019). This rheological contrast, which is presented at a range of scales, will strongly enhance intrasalt deformation and development of buckle folding or boudinage structures, and at high strains can also lead to a tectonic melange (Raith et al., 2016, 2017). However, in relatively pure rocksalt, extrapolation of laboratory creep rates in rocksalt to low stress-low strain rate conditions is not well known, and even less is known about the effect of impurities on this low stress-low strain

rate rheology.

In this paper, we investigate rheological variation of folded rocksalt based on the analysis of the buckle fold geometry. Shape of buckle folds is a sensitive parameter to rheological properties of the layers and various studies show suitability of these structures in deciphering rheology of various rocks (for review see: Hudleston and Treagus, 2010; Schmalholz and Mancktelow, 2016). The Ocnele Mari salt mine, located in the Southern Carpathians of Romania, exposes spectacular

structures on the cleaned walls of over 50 regularly arranged pillars, ceiling and floor. We focus on the analysis of folds that develop on various scales, which are often referred to as polyharmonic folds. Development of the polyharmonic folds is limited





to a specific combination of geometrical and rheological parameters of the multilayer sequence (e.g., Treagus and Fletcher, 2009). As a result, these structures are potentially useful in deciphering mechanical variation between the layers.

Field observations from Ocnele Mari salt mine are used as an input for numerical models to constrain range of geometrical parameters of the multilayer sequence. Moreover, microstructural analysis provides support for the assumption that the rheology of rocksalt was linear viscous during the folding. We employ numerical simulations to study the influence of viscosity ratio between the layers on the developing structures for slightly varying initial model setups. We show that the polyharmonic folds develop only for a selected range of values. Detailed comparison between the field observations and numerical results suggest that the viscosity ratio between the layers varied between 10 and 20.

**2    Regional Setting in Carpathian Geology**

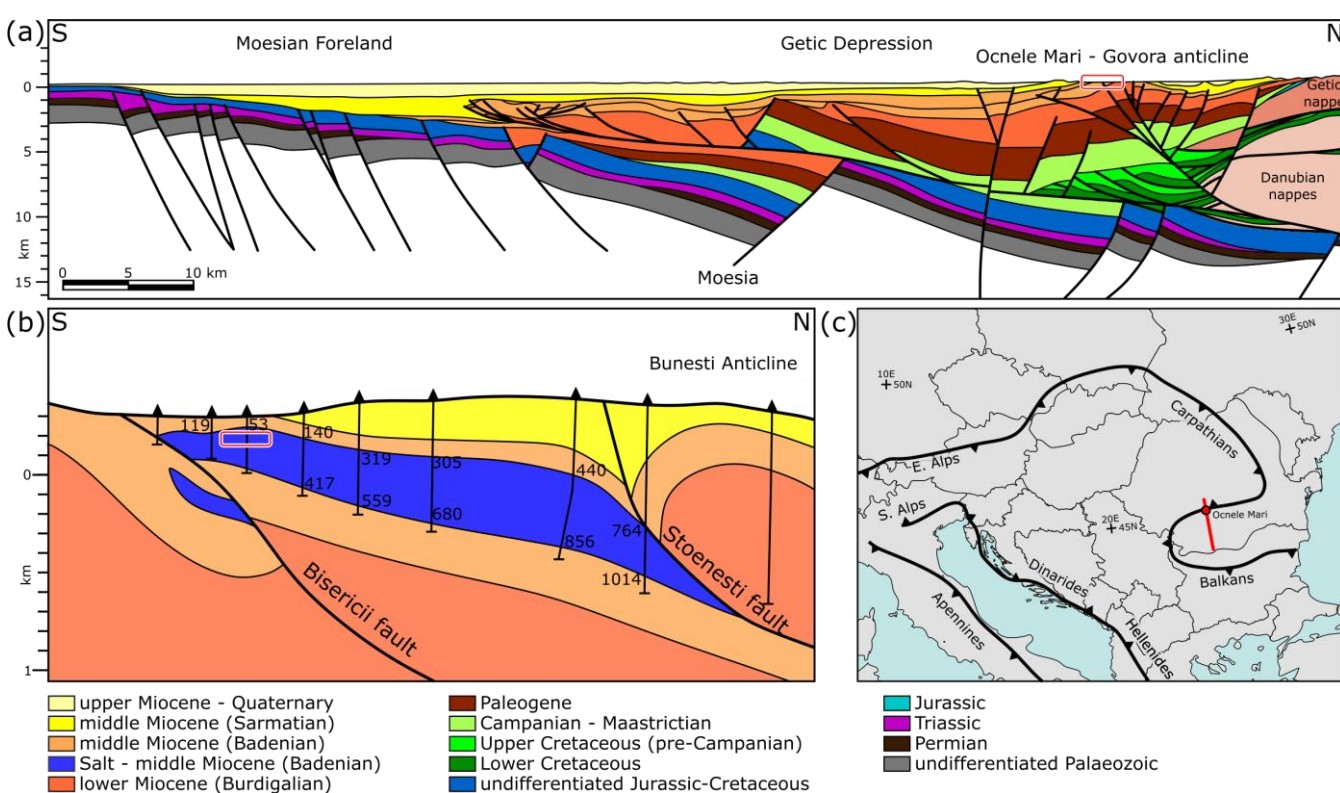

**Figure 1. (a) Regional geological profile of the area (after Răbăgia et al., 2011 and references therein). The location of the local profile (b) is marked with red rectangle. Coordinates of the two ends of the profile are: north 45°20'51"N 24°19'51"E south 43°47'11"N 24°31'43"E. (b) Local geological profile of the Ocnele Mari area, illustrating the shape and position of the salt body**
**(45°05'06.9"N 24°18'33.7"E). (after Stoica and Gherasie, 1981) with location of wells, on which the profile is based. The numbers indicate top and bottom of salt (in measured depth). Approximate location of the mine is marked with red rectangle. (c) Sketch of main structural features of the Alpino-Carpatho-Dinaric region. Location of the regional profile (a) is marked with a red line.**



The study area is located in the Romanian South Carpathians, in the thin-skinned part of this fold and thrust belt (Fig. 1a). The Carpathians are an Alpine orogen that records the late Jurassic to middle Miocene closure of the Alpine Tethys ocean (Săndulescu, 1988, 1984; Csontos and Vörös, 2004; Schmid et al., 2008; Maţenco, 2017). This thin-skinned mountain belt is located at the contact between the South Carpathians and the Moesian platform (Fig. 1a, c) is commonly known as the Getic depression (Motaş, 1983; Răbăgia et al., 2011; Krézsek et al., 2013). The sediments of the Getic depression range from latest Cretaceous to Quaternary and were deposited as post-tectonic to the Cretaceous thick-skinned deformation of the South Carpathians (Rabăgia and Maţenco, 1999; Răbăgia et al., 2011; Krézsek et al., 2013). The evolution of the Getic depression is characterised by transtensional opening during the Paleogene-early Miocene period, followed by south-directed thrusting, and transpression (inversion) during Middle Miocene to Quaternary times (Krézsek et al., 2013).

The Ocnele Mari salt mine is located in the Govora-Ocnele Mari antiform (e.g., Răbăgia et al., 2011; Krézsek et al., 2013). It evolved as an early Miocene extensional structure that was repeatedly inverted by subsequent phases of shortening (Răbăgia et al., 2011; Krézsek et al., 2013). There are two main areas with structural highs in this antiform, the Govora anticline (west) and the Ocnele Mari anticline (east) with multiple tear faults striking from NNE-SSW to NNW-SSE (Popescu, 1954; Răbăgia et al., 2011).

The Ocnele Mari salt body is approximately 10 km long and 3.5 km wide (Stoica and Gherasie, 1981; Zamfirescu et al., 2007) and it is Middle Miocene in age (Popescu, 1954; Iorgulescu et al., 1962; Murgoci, 1905). The salt body is dipping 15-20° to the north and is over 400 meters thick in its central part as defined by well and mining data (Fig.1 b; Zamfirescu et al., 2007). It is likely that the deposition of salt was laterally uneven, controlled by the pre-mid-Miocene topography (Fig. 1b, Popescu, 1954), before it became one of the evaporite detachments of this fold and thrust belt. Most of the salt is more than 97% NaCl with an alternation of lighter (white) and darker layers (Stoica and Gherasie, 1981). Rare micro fossils include nuts, chestnuts, pinecones, and charred coal fragments (Iorgulescu et al., 1962). Due to the complex tectonic history and the poor fossil record in the Lower to Middle Miocene formations, there are still large uncertainties regarding the age of some salt formations today (see Filipescu et al., 2020).

## 3  Ocnele Mari salt mine

The Ocnele Mari salt mine is one of the many salt mines in Romania that is open to the public. It is located in the South Carpathian region in the upper, near-surface part of the Ocnele Mari salt body (45°05'06.9"N 24°18'33.7"E). Mining activities in the Ocnele Mari region were ongoing since Roman times (11th to 13th century) (Stoica and Gherasie, 1981; Tămaş et al., 2021a). Salt from this area is being exploited with both solution mining and room and pillar mining. For more details on the location of the mine, other such exposures, and the history of salt tectonics in the Romanian Carpathians, see Tămaş et al. (2018, 2021a). The active salt mine is being exploited at two distinct levels (+226 m and +210 m above sea level), one of which is accessible to the public (horizon 226, ~50 m below the surface), exposing spectacularly folded rocksalt.



## 4    Multilayer buckle fold analysis

Multilayer buckle folds show a great variety of shapes, related to the large number of parameters that influence the folding process. The most important factors are: 1) model geometry e.g., number of layers and their thickness, geometry of the layer interfaces (type of perturbation and its amplitude), 2) mechanical properties of the layers e.g., viscosity ratio, stress exponent for the power-law flow law, anisotropy, 3) contact between the layers i.e., no-slip (bonded) or free-slip, 4) amount of shortening, and 5) boundary condition e.g. free-slip, no-slip, free-surface, rate of deformation.

Understanding of folding in multilayers is rooted in the analysis of deformation of an isolated more viscous layer embedded in the less viscous matrix. Biot (1961) and (Ramberg, 1962) described the wavelength selection process, which is responsible for the faster growth of selected wavelengths and leads to development of a semi-regular pattern of the fold train. The wavelength that initially experiences the largest growth rate is referred to as dominant wavelength. Dominant wavelength normalized by the layer thickness is a function of the viscosity ratio between the layer and matrix. Layers with larger viscosity

ratio tend to develop larger dominant wavelengths and grow faster. Further studies on the wavelength selection process allowed to establish a relation between the single-layer fold shape and the rheological parameters (Biot, 1961; Sherwin and Chapple, 1968; Fletcher and Sherwin, 1978; Schmalholz and Podladchikov, 2001).

Numerous theoretical (e.g., Biot, 1961, 1965; Johnson and Fletcher, 1994), analogue (e.g., Ghosh, 1968; Cobbold et al., 1971; Ramberg and Strömgård, 1971), and numerical (e.g., Frehner and Schmalholz, 2006; Schmid and Podladchikov, 2006;

Schmalholz and Schmid, 2012) studies aimed to investigate multilayer buckle folding and the wavelength selection process. The studies show that folding instability increases with an increasing: 1) number of stiff layers, 2) viscosity ratio between the stiff and soft layers, and 3) thickness of the bounding soft layers (e.g., Johnson and Fletcher, 1994). However, establishing a general relation between the parameters that control multilayer folding and the resulting fold shape is strongly hampered due multiple controlling factors and associated non-unique solution. Additional difficulty of the analysis involves determining fold

shape parameters that can accurately describe the often complex multilayer pattern.

In various studies, evolution of the multilayer structures is limited to the analysis of a specific configuration of the layers. Johnson and Pfaff (1989) examined conditions of development of parallel, similar and constrained folds in multilayers composed of alternating stiff and soft layers with bonded or free-slip contact or multilayers composed of all stiff layers with free-slip contact. The authors illustrated an example of how these structures can be used to infer viscosity ratio between the

layers and also conditions of deformation. The role of spacing between the layers was discussed by Ramberg (1962), who distinguished disharmonic and harmonic folds. He showed that if the separation between the layers is larger than the sum of their dominant wavelength, the layers fold independently, forming disharmonic structures, otherwise they start to interact and fold harmonically. Decreasing distance between the layers causes them to behave as an effective single layer. Theoretical and numerical study of the transition between single, multilayer and effective single layer deformation was presented by Schmid

and Podladchikov (2006). In the cases, when the multilayer sequence consists of layers that interact with each other and have strongly variable thickness and/or variable rheology, folds can grow on various scales forming a polyharmonic structure.



Conditions of when the polyharmonic structures can develop in the multilayer sequence was investigated by Frehner and Schmalholz (2006) and Treagus and Fletcher (2009).

Treagus and Fletcher (2009) suggested that small-scale structures that develop in the thin layers, in order to survive, must have a greater growth rate than the large-scale structures. However, growth rate of the multilayer package is always larger than that of the single layer. The authors concluded that the variation of the layer thickness alone in various cases is insufficient to generate polyharmonic folding. Yet, two factors can reverse this relation: (i) increasing viscosity of the thin layer strengthens amplification of the small-scale structure and (ii) confinement of the multilayer stack (i.e., confined between two parallel rigid planes) suppresses amplification of the large-scale folds.

Frehner and Schmalholz (2006) tested numerically the development of polyharmonic folds in a multilayer package containing alternating stiff and soft layers with variable thickness. In their models, the initial perturbation amplitude was equal for thin and thick layers causing relatively greater interface roughness of the thin layer. The authors argue that using the same perturbation amplitude for all the layers could be a reasonable assumption in various rock sequences in particular in sedimentary rocks. Consequently, the small-scale folds developed two wavelengths as they achieved large amplitudes before they were incorporated into large-scale fold structure.

## 5    Buckle folding in simple shear

Most of the theoretical, analogue, and numerical studies related to buckle folds in multilayers concentrate on the analysis, where the principal direction of shortening is parallel to the layering (e.g., Biot, 1961; Currie et al., 1962; Biot, 1965; Frehner and Schmalholz, 2006; Frehner and Schmid, 2016). Only few studies analyse folding of multilayer when the shortening direction is oblique to the layering (Cobbold et al., 1971; Ghosh, 1968; Schmalholz and Schmid, 2012). Results of numerical analysis of single layer folding in pure and simple shear presented by Llorens et al. (2013) demonstrate no large difference between folds that develop in these two regimes. As pointed by R.K. Davis and R.C. Fletcher (personal communication), this is due to the fact that folding of linear viscous layers, to the first order, depends only on the bulk layer-parallel shortening.

## 6    Methods

### 6.1    Field and photogrammetry techniques

Several fieldwork campaigns were carried out in the Ocnele Mari salt mine with the scope of structural mapping, acquiring high-resolution photographs of all the accessible pillar walls and part of the mine ceiling (~3300 photographs) and measuring the orientation of the observed structural features. Key photographs within the mine were selected and interpreted using vector graphics software (Inkscape).

To aid the 3D structural analysis, 3D digital outcrop models and orthorectified models have been created using Structure from Motion (SfM) photogrammetry techniques. Agisoft Metashape Professional (v.1.6.2) was used for the creation of these models





and Virtual Reality Geological Studio (VRGS v.2.52) software (Hodgetts et al., 2007) was used for interpreting them (for more details on outcrop creation and interpretation (see Tămaș et al., 2021b).

In addition to data extracted from the 3D digital outcrop models, we measured the orientation of salt foliations and fold axes

in the mine, using a traditional Freiberg geological compass and with an iPad 9.7 2018 (using FieldMove software). Orientation data were processed in Structural Geology to PostScript (SG2PS; Sasvári and Baharev, 2014). The measurement of these structural features was allowed by the geometry of the pillars, as foliation and fold axes could be traced "around the corner" to the other side of the pillar.

## 6.2     Numerical modelling

The open-source software FOLDER, which is designed for the analysis of the deformation in layered medium in two-dimensions (Adamuszek et al., 2016), was used to model evolution of the multilayer fold structures. FOLDER builds on MILAMIN (Dabrowski et al., 2008) to solve Stokes equations for incompressible viscous flow under zero gravity using the Finite Element Method. A high quality unstructured mesh is created using a triangular mesh generator Triangle (Shewchuk, 1996). Moreover, FOLDER includes a range of utilities from the MUTILS package (Krotkiewski and Dabrowski, 2013), which

allow for high resolution modelling. The simulations were conducted using the Neptun computational cluster in the Polish Geological Institute - NRI.

## 6.3     Fold shape analysis

Fold shape of a single layer fold example is analysed using Fold Geometry Toolbox, FGT (Adamuszek et al., 2011), where the following definitions of the parameters are used. Arclength is determined along the interface as the distance between two

neighbouring inflection points. Amplitude is measured as the distance between the line joining two inflection points and the extremity of the fold, whereas wavelength is twice the distance between two inflection points (Ramsay and Huber, 1987). For layer thickness, we use a mean thickness value calculated as the area of the layer divided by the average arclength (Adamuszek et al., 2011). Finally, for the example studied here, we estimate viscosity ratio between the layer and embedding matrix using Fletcher and Sherwin (1978) and Schmalholz and Podladchikov (2001) methods.

## 6.4     Microstructural analysis

The microstructural analysis of regular and Gamma-irradiated thin sections was performed based on two hand specimens collected in the Ocnele Mari mine: RO-OM-01 and RO-OM-02. The samples were cut perpendicular to the bedding in a dry laboratory with a diamond saw and cooled by a small amount of slightly undersaturated salt brine to reduce mechanical damage. Regular thin sections were dry polished to a thickness of approximately 1 mm and Gamma-irradiated thin sections

were dry polished to a thickness of approximately 50 µm to allow for decorated microstructures to be visible. To create a negative topography at grain boundaries and subgrain boundaries, the samples were chemically etched with slightly undersaturated brine and flushed with a stream of n-hexane using the technique described in (Urai et al., 1987). The thin



sections were imaged in reflected and transmitted light on a Zeiss optical microscope with stitching panorama function of the ZEN imaging software. To decorate crystal defect structures, samples were irradiated in the research reactor FRM-II at the TU

Munich in Garching with varying dose rates between 6 kGy/h and 11 kGy/h to a total dose of 4 MGy at constant temperature of 100 °C (Urai et al., 1986; Schléder and Urai, 2005; Schléder et al., 2007).

For X-ray diffraction (XRD) analysis of inclusions, part of RO-OM-02 was dissolved in deionized water and the insoluble residue vacuum filtered. Qualitative and quantitative XRD measurements were then performed on a Bruker D8 equipped with a graphite monochromator and a scintillation counter. Scans were measured with Cu-kα radiation.

Vitrinite reflance measurements were performed on RO-OM-01, to obtain information on the maximum temperature reached during burial. Due to the low organic content of the sample, the sample was crushed by hand to a grain size of about 1 mm ensuring no vitrinites get destroyed. The halite fraction was dissolved by several washing steps with distilled water until only insoluble components remained. After drying, this residue was then impregnated in a two compound epoxy resin and polished. Reflectance values were measured on a Zeiss Axioplan microscope equipped with a 50x/1.0 EC-Epiplan-NEOFLUAR Oil Pol

objective lens and a PI 10x/23 ocular lens. Images were recorded using a Basler CCD camera and image processing was performed using the Fossil software (Hilgers Technisches Büro). Calibration of reflected incident light intensity was performed with a Leuco-Saphir standard with a reflectance of 0.592%. 50 individual vitrinite grains were measured to calculate the mean random vitrinite reflectance (VRr). VRr was translated to the maximum past temperature using the equation by Barker and Pawlewicz (1994) for burial heating.

Halite grain and subgrain boundaries were manually traced with a touchpen and tablet and analysed with Fiji (Schindelin et al., 2012). Subgrain size for piezometry was calculated as equivalent circular diameter (Schléder and Urai, 2005).





# 7     Results

## 7.1     Overview of the mine

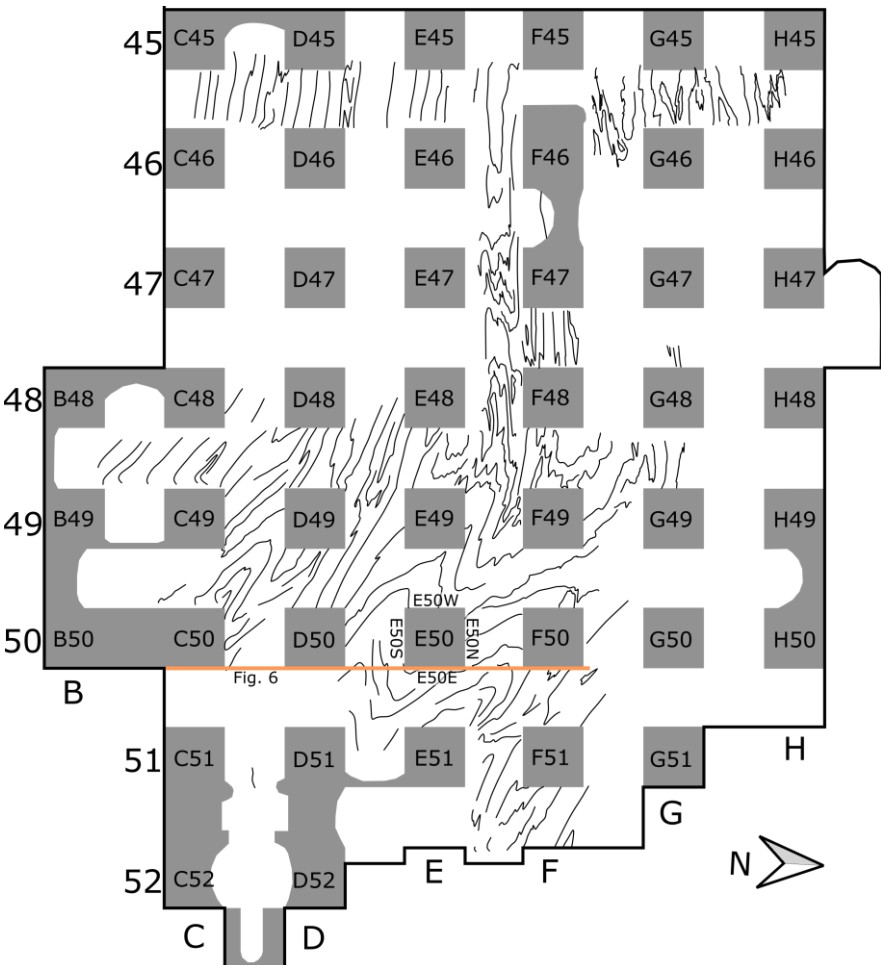

**Figure 2. Map of the studied area in the +226 horizon of the Ocnele Mari salt mine, illustrating position of the pillars. Curves between the pillars show traces of the layers observed on the mine ceiling. Thick orange line along the eastern sides of the line 50 marks position of the cross-section through the mine. The public area of the mine (shown in the map) is approximately 180 m by 210 m and has a height of approximately 8 meters.**

The public area of the mine (studied area) is approximately 180 m by 210 m and has a height of approximately 8 meters. The galleries are oriented approximately N-S and E-W separating over 50 square pillars with an edge length and spacing between the pillars of around 15 meters (Fig. 2). The structures in the mine can be observed on the clean surface of pillars, walls and ceiling allowing for the detailed three-dimensional analysis. Figure 3 illustrates a three-dimensional digital model of pillar D49.





Figure 3. (a) 3D digital model of pillar D49 illustrating the NE-SW fold axis. (b) Photograph of the east-facing pillar wall. (c) Photograph of the north-facing pillar wall. (d) Photograph of the west-facing pillar wall. (e) Photograph of the south-facing pillar wall.






**Figure 4. Photos of the eastern and western sides of the pillar C50 illustrating different scales folding and the range of fold shapes. Note that the layers in the fold limbs are locally boudinaged. See text for further discussion.**

The salt exposed in the studied area of the Ocnele Mari salt mine typically shows an alternation of darker and lighter layers of rocksalt (Figs. 3, 4, and 5). The light-coloured layers are commonly multilayered packages, centimeter to decimeter thick and





consist almost entirely of a granular aggregate of halite crystals, while the darker layers are multilayers consisting of thinner
(millimeter to centimeter thick) layers in many shades of grey (Fig. 4b, f). Together with these darker and lighter salt layers, in many cases, locally we find thin, millimeter to centimeter-scale layers rich in clastic material (shale, silt and sand) or gypsum and anhydrite-rich layers. These inclusion-rich layers are often either black in colour or yellow-brown (the silty-sandy ones) (Fig. 4d).

Folds are the most impressive structures in the Ocnele Mari salt mine and can be observed on most of the east- or west-facing
pillar walls. Folds occur on various scales ranging from centimetres to tens of meters and represent a variety of geometries including harmonic, polyharmonic and disharmonic styles.

Figure 4a shows an example of the hinge region of the large-scale multilayer fold structure in the eastern wall of pillar C50. The large-scale fold is tight and angular. On the limbs, smaller-scale asymmetric folds commonly occur forming a characteristic Z- and S-shape. More complex geometry is observed in the hinge of the fold, where different folds develop
depending on the distance between dark layers. The black layer in Figs. 4b, embedded in a relatively thick light-colour (white and bright grey) beds locally develops geometry characteristic for a large amplitude single layer buckle fold (indicated with a black arrow in Fig. 4b). Close spacing between the black and yellow-brown layers in Figs. 4c leads to development of the two multilayer packages that behave as effectively two layers. These "two-layers" form 10-40 cm scale folds with close to tight geometry and subrounded to angular hinges. Note, that locally the very thin yellow-brown layers form smaller scale folds
(indicated with a black arrow in Fig. 4c). Bright layer separating the "two-layers" is characterized with a considerable thickness variation as compared to the other "layers". Fold shapes with angular hinges shown in Fig. 4d develop in the multilayer package, where the ratio between spacing between the black and yellow-brown layers and their thickness is ca. 1. Note the folded yellow-brown layer within the black layer (indicated with a black arrow in Figs. 4d).

Locally, thin-layer greyish bands within the white material (in the top of Fig. 4c) form folds with angular to sharp hinges. The
limbs of the folds are characterized by a wavy shape that follows the shape of the neighbouring layers. The shape of these folds clearly differs from the other examples of folded layers.

Figure 5a and c illustrate other examples of the polyharmonic folds that develop in the hinge region of the larger-scale folds. Similarly as in Fig. 4, depending on the thickness variation of the individual layers and the spacing between the layers, we observe folds variable geometry of the fold shapes developing on different scales.

The style of the multilayer folding can change along the layer, which is often a result of the lateral variation of the layer thickness. Moreover, some of the layers are boudinaged (indicated with white arrows in Fig. 4b,d), whereas others have locally diffuse boundary (Figs. 4e, 5b).

Refolded folds have been observed in several pillars, floor and ceiling (Fig. 5e,f). These are not common and they only locally affect the morphology of the folds discussed here, but the examples are consistent with strong deformation before the folding.







Figure 5. Fold structures on: (a,b) western-facing wall of pillar F48, (c,d) eastern-facing wall of pillar E50, and (e,f) eastern-facing wall of pillar D50E. See text for further discussion.





## 7.2 Polyharmonic folds

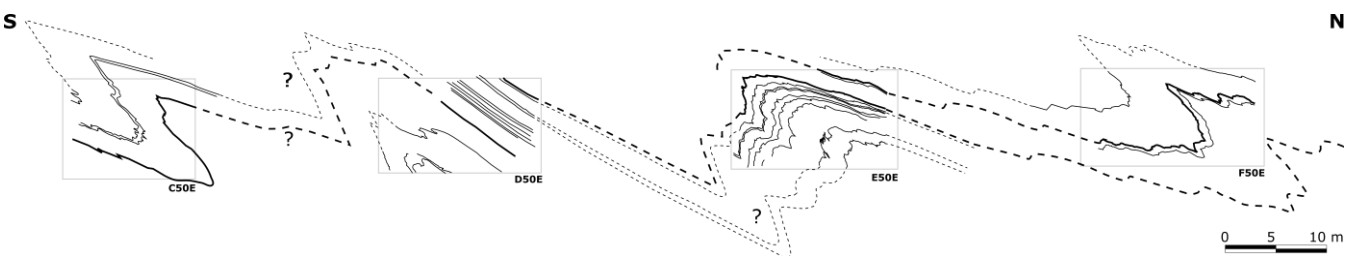

**Figure 6. Cross-section across the mine along the eastern side of pillar line 50 (see Fig. 2 for overview). Rectangular boxes illustrate the position of pillars. Dashed lines indicate correlations between pillars, based on layer morphology and on observation on the ceiling and floor and on the other sides of the pillars.**

To illustrate the geometry of folds at the scale of the mine, we constructed a N-S cross-section through the mine (Fig. 6) along the 50 line (see Fig. 2). Between the pillars, layers were correlated by (i) similarities in the multilayer stratigraphy (keeping in mind that layer thickness can vary strongly between fold hinge and fold limb, and that not all layers in one pillar are laterally continuous, and (2) by tracing layers from one pillar to the next, along the ceiling and floor (see Fig. 2). The largest-scale folds generally have straight limbs and angular hinges. The folds shapes range from open to isoclinal. They are strongly asymmetric, where the short limb is ca. 10-20 m long, whereas the long limb 30-60 m long. The folds are verging south and dip to the N, consistent with top-to the south shear in a salt detachment. The fold axes are near-horizontal. E-W in the W part of the mine, and NW-SE in the East (Fig. 7a). The orientation of the foliation changes from north-dipping in the western side of the salt mine (Fig. 7b; pillar lines 45-48) to north-east dipping in the eastern side of the salt mine (Fig. 7c; pillar lines 48-52).

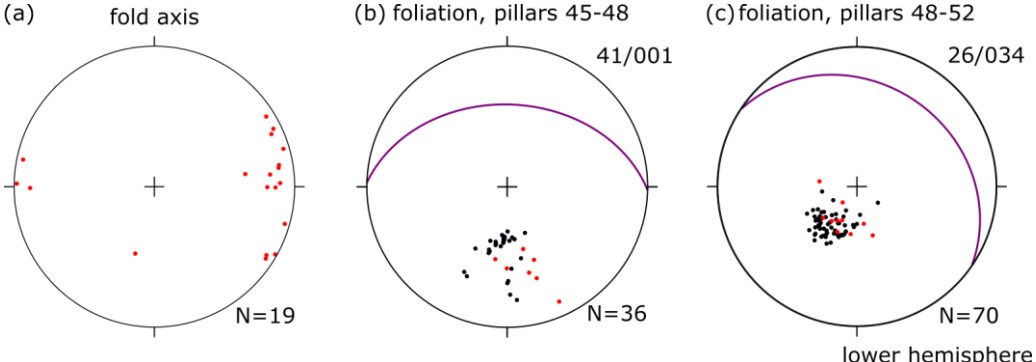

**Figure 7. Lower-hemisphere stereonet plots showing (a) fold axes (all measured with the compass) and (b,c) poles to measured foliations and average orientation as a great circle. Black poles resulted from the interpretation of the 3D digital models of the mine, while the red poles were measured in the field.**







**Figure 8. (a) Polyharmonic fold structures observed on the eastern-facing wall of pillar E50 (Fig. 4d), used for the numerical modelling. Black lines trace the layer interfaces, which illustrate from the bottom large-, middle, and small-scale fold structures. (b) and (c) show zoom-in of the middle- and small-scale fold structures. Note that the photo is rotated and the arrow in (a) points upwards.**






We select an excellent, representative exposure of the polyharmonic folds in the multilayer package from the eastern-facing wall of pillar E50 for a detailed fold shape analysis (see Fig. 2a). We rotate the image, so the axial planes of the large-scale fold structure are vertical and indicate the top direction with the arrow (c.f. Fig. 4d and Fig. 8). The package consists of

interbedded dark and white layers of different thicknesses. In the sequence, we distinguish 5 domains, which comprise a set of closely spaced layers and refer to them as D1-D5 (see Fig. 8a). Most of the layers are dark grey in colour and only locally yellow-brown layers are found.

The multilayer sequence exhibits a multiple order of folding (2nd, 3rd, and 4th), which we further refer to as large-, middle-, and small-scale, respectively. The structures in the pillar are located in the hinge area of the largest 1st order fold structure that

we have identified on the mine-scale (see Fig. 6). In Fig. 8a, three selected interfaces (marked with black curves) in the D1, D2, and D4 domains illustrate large-, middle, and small-scale folds, respectively.

All dark layers are characterized by a less variable thickness compared to the white layers and, thus, are considered to have higher viscosity (e.g., see white layers embedding D4 domain). The minimum layer shortening estimated based on the relation between the layer arclength and layer length varies between 35-45% depending on the layer.

All the layers within the package exhibit a common, harmonic geometry with approximately similar wavelength and amplitude. The fold wavelength varies between 60 and 200 cm. The large-scale folds characterize the fold pattern of the relatively thick D1 and D5 domains, which, due to small thickness of the internal white layers, appear as massive units controlling the large-scale deformation. The folds are reasonably symmetric in the hinge regions. The interlimb angles vary around 90° and folds can be classified as open (according to Fleuty, 1964). Locally, where the individual dark layers are embedded in a thicker

white salt, we can observe even smaller scale fold structures developing within the domains.

Middle-scale folds are visible in the D2 and D3 domains. Both domains are composed of white and dark layers of approximately similar thicknesses. Folds that developed on the long limbs of higher-order folds are slightly asymmetric and show typical 'Z' and 'S' geometry (Fig. 8b). The folds show gentle to open geometry with rounded hinges. Typical fold wavelength varies between 15 and 30 cm.

The small-scale folds reveal wavelengths of 1-5 cm scale and are characteristic of the D4 domain. The domain contains two thicker, lower, and thinner, upper, dark layers interbedded with a white layer (Fig. 8c). The lower layer is ca. 1 cm thick and shows clear folding pattern. Folds have variable open to tight geometry with no significant shape asymmetry. The thin layer has less than 0.2 cm in thickness and generally illustrates low amplitude, gentle folds. D4 is embedded in the relatively thick white layers. The whole domain follows the large-scale fold geometry, whereas we do not observe clear undulations

characteristic for the middle-scale wavelength.





## 7.3    Fold shape analysis

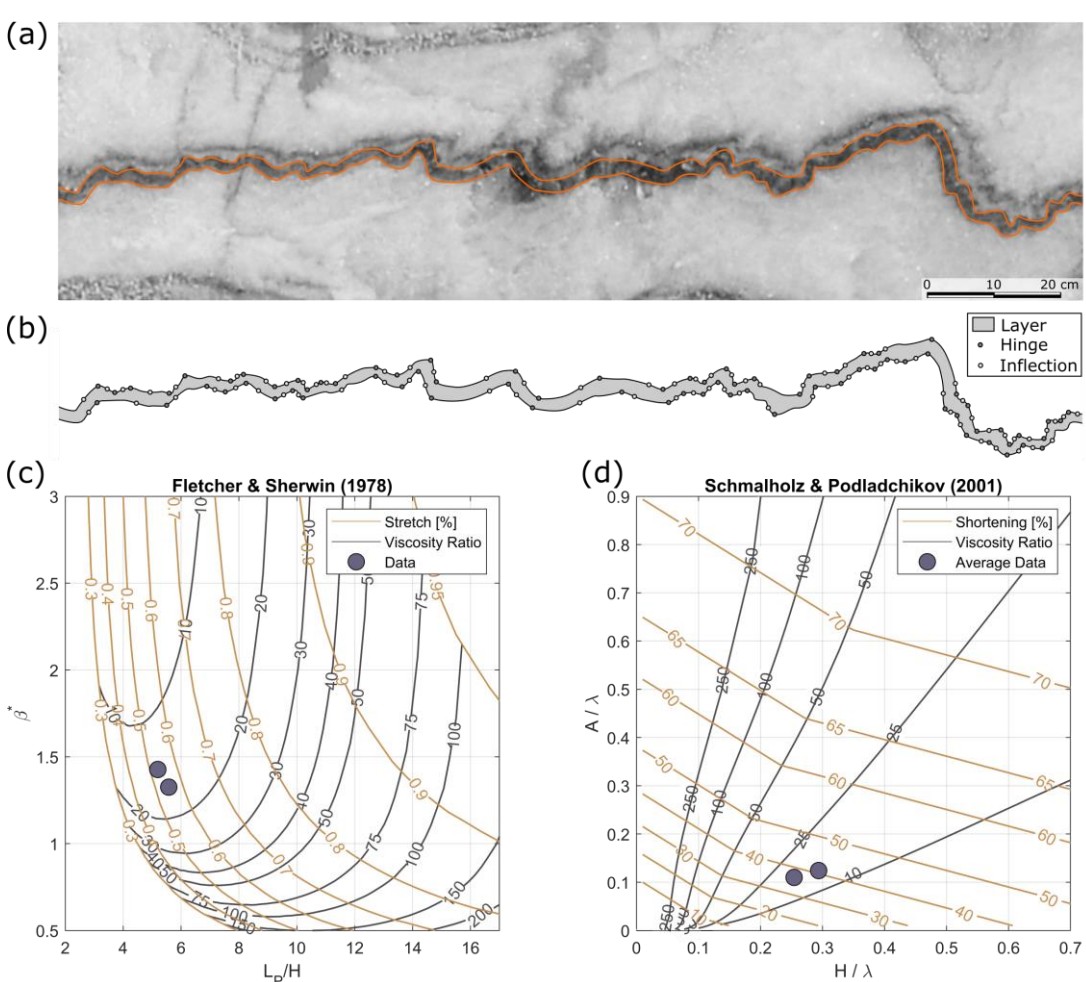

**Figure 9** Fold shape analysis using FGT. **(a)** Small-scale fold structure. **(b)** Position of inflection points and hinges. **(c)** Estimates of layer-parallel shortening stretch and viscosity ratio after Fletcher and Sherwin (1978). **(d)** Estimates of shortening and viscosity ratio after Schmalholz and Podladchikov (2001).

Single dark-layer folds surrounded by a thick white layer are rare in Ocnele Mari. However, a selected fragment of the bottom dark layer in the D4 domain is used to estimate the viscosity ratio between the layers using FGT (Fig. 9a). Figure 9b shows positions of hinges and inflection points determined by the toolbox, whereas Fig. 9c-d illustrates the diagram of the viscosity ratio estimates by Fletcher and Sherwin (1978) and Schmalholz and Podladchikov (2001) methods, respectively. Two dark dots in the plots indicate the analysis for the two interfaces. The method by Fletcher and Sherwin (1978) yields a viscosity ratio in a range between 10 and 20. Stretch, at which the wavelength selection took place, is ca. 0.55. The viscosity ratio estimated using Schmalholz and Podladchikov (2001) for mean values of fold amplitude to wavelength ratio (A/λ) and mean thickness to wavelength ratio (H/λ) ranges between 10 and 25. Amount of total shortening is around 40%.





## 7.4     Microstructure

**Figure 10. (a) Hand specimen RO-OM-01 with photomontaged transmitted light image of thin section shows white salt in the matrix and folded multilayers in the centre surrounded by dark salt. (b) Transmitted light image of Gamma-irradiated thin section of**



**sample RO-OM-01 with blue decoration of halite microstructures and traced halite grain boundaries. Folded salt with elongated claystone fragments is highlighted in green. Numbering (1 – 18) indicates halite grains that were used for subgrain size piezometry**
**(reflected light micrographs and measurements are provided in Supplements S1.1and S1.2). (c) Transmitted light micrograph of fold hinge showing subgrains in Gamma-decorated halite and aligned cubic fluid inclusion bands following the orientation of the fold, location is indicated in (b). (d) Transmitted light micrograph of white salt showing elongated halite grains rich in subgrains and overgrowth rims, location is indicated in (b). (e) Transmitted light micrograph of halite showing decorated white subgrains and impurities and fluid films at halite grain boundaries, location is indicated in (b). Reflected light micrograph of image (e) showing**
**etched subgrains in halite grain with darker reflectance and also subgrain free halite grains, location is indicated in (b).**

Microstructural investigations of the Ocnele Mari salt were performed based on two hand specimens that show different yet individually rich rock fabrics. The unoriented samples were recently mined and were collected at the entrance of the tourist area. The samples are representative of the white, grey and beige bands present in the pillars (Fig. 10a,b), and contain one of the thin, darkest layers shown in Figs. 4 and 5. Sample RO-OM-01 is a small fold, representing the hinge areas discussed above
(sectioned perpendicular to the fold axis), and RO-OM-02 is a sample with straight foliation, inferred to represent the limbs of the folds studied.

### 7.4.1    White salt

The white salt in both samples is clear halite, with elongated halite crystals with an average grain size of 2.1 mm (larger grains of up to 10 mm and an average aspect ratio of 2.8 in sample RO-OM-01. Digitized grain boundaries and inclusions of sample
RO-OM-02 as well as grain size measurements are provided in Supplements S2.1 and S2.2. In sample RO-OM-02, halite grains of the white salt around a large anhydrite inclusion are smaller, with an average grain size of 1.6 mm. In sample RO-OM-01, the grains' long axes are parallel to the axial plane in the fold cores and parallel to the folded layer outside of the fold (Fig. 10b). In sample RO-OM-02, the orientations of the grains' long axis are sub-parallel to the layering (Fig. 11b). The grain boundaries in the white salt are mostly straight or slightly curved, with locally lobate morphologies (Figs. 10e, 11c,e).
Gamma-decoration of white salts in sample RO-OM-01 shows mostly light or dark blue grain cores with white growth bands, which are locally truncated (Fig. 10b,d). Light grain cores with pale blue rims are also present (Fig. 10d). These can contain cleavage cracks, which are filled with fluid inclusions and have a pale blue Gamma-decorated halo of approximately 80μm radius. Some grains show decoration of weak slip bands. Subgrains decorated by Gamma-irradiation can appear as both; either blue subgrain boundaries in white halite grains or white subgrains in blue halite grains. The white salt halite grains in sample
RO-OM-02 usually has dark blue cores with Gamma-decorated subgrain boundaries and white rims. Subgrain boundaries appear white in blue halite cores. It is interesting to note that in both samples, not all subgrains boundaries marked by Gamma-decoration appear on the etched surfaces under reflected light (Fig. 10e). This is surprising because EBSD studies have shown that the chemical etching procedure is very sensitive and decorate subgrains with even very small misorientations, (Trimby et al., 2000) and needs further study. Only the very bright subgrain boundaries visible under transmitted light are also consistently
present under reflected light. Grain boundaries show abundant fluid inclusion arrays (Fig. 10d).





### 7.4.2    Dark salt

The dark salt of both samples RO-OM-01 and RO-OM-02 is rich in second phase inclusions and consists mostly of halite with elongated aggregates of clay that have an average aspect ratio of 4 and rounded particles of anhydrite with an aspect ratio of 2 (Fig. 10a, b, 11b, S2.1 and S2.2 in Supplements). Fluid inclusions at grain boundaries and ghost grain boundaries are abundant
(Fig. 10f, Fig. 11f). Thin layers of second phase inclusions are commonly boudinaged in both samples (Fig. 10a, c and Fig. 11c) folded (Fig. 10a, b) or bent (Fig. 11b). Anhydrite grains in these thin layers are intergrown with the halite matrix (Fig. 10a, 11a, b, e, f). The boudin necks contain fibrous halite crystals (Fig. 11c and e) (c.f. Leitner et al., 2011). The grey salt halite grain size of 1.3 mm in sample RO-OM-02, smaller than in the white salt (2.1 mm): the inclusion-rich salt is consistently finer grained than the pure salt (Krabbendam et al., 2003) (Supplement S2.1 and S2.2). Sample RO-OM-01 has abundant fluid
inclusions inside halite crystals of the dark folded layer that are aligned and parallel to the folded layer. These aligned fluid inclusions are partly overgrown with halite crystals that contain subgrains visible through Gamma-decoration (Fig. 10c). We observed two extremely thin "layers" of impure salt outside the fold, following the fold shape (Fig. 10a).

X-Ray diffraction analysis of insoluble residue was measured in sample RO-OM-02 (Fig. 11a). Results show the presence of anhydrite and gypsum (combined 39 wt.%), clay minerals smectite, muscovite-illite and chlorite (combined 34 wt.%), and
minor quarz, orthoclase, albite, dolomite, apatite and calcite. Individual measurement of larger white particles indicated them to be mostly anhydrite (90 wt.%) so that we interpret the presence of gypsum as a consequence of anhydrite hydration in water during dissolution in the preparation process. Further, the analysed sample contains 6 wt.% amorphous phases that are interpreted to be organic material. This is supported by the findings of vitrinite within claystone fragments on thin sections under reflected light microscopy. Measurements of vitrinite reflectance indicates a maximum heating temperature of 58°C
according to Barker and Pawlewicz (1994).

### 7.4.3    Subgrains

Gamma-decoration illuminates abundant presence of subgrains that are mostly visible as blue patches with white subgrain boundaries or white subgrains with delicate blue subgrain boundaries (Fig. 10d and 11d). However, these abundant subgrains are either weakly or not resolved on the etched surface under reflected light. Only a few halite grains show well-defined
subgrain boundaries, often these grains have lower reflection (Fig. 10f, S 1.1 in Supplements) and under transmitted light, these subgrain boundaries appear very bright (Fig. 10e). Subgrain size piezometry (Schléder and Urai, 2005) of RO-OM-1 indicates variable mean subgrain sizes for individually measured halite grains. The halite grains with smallest subgrains show high differential stresses such as 4 MPa as in the case of grain 11 (in Fig, 10b, e and f), with a logarithmic mean subgrain size of 1.65µm, n=187. Measurements and micrographs of further grains are provided in the Supplements. Grains with larger
subgrains such as grain 1 and grain 10 (Fig. 10b) indicate low differential stresses of about 0.4 MPa according to Schléder and Urai (2005). Based on the average subgrain size of all measured grains with subgrains, a differential stress of 2.3 MPa for sample RO-OM-01 was calculated.







**Figure 11. (a) Hand specimen RO-OM-02 with photomontaged transmitted light image of thin section shows bands of white and dark salt with elongated fragments parallel to the foliation. (b) Transmitted light image of Gamma-irradiated thin section of sample RO-OM-02 with blue decoration of halite microstructures and traced halite grain boundaries. Elongated claystone fragments, anhydrite and salt layer boundaries are indicated. (c) Transmitted light micrograph shows boudinaged claystone aggregate with**





thin layers of halite and fiborous halite in the boudin neck, location is indicated in (b). (d) Transmitted light micrograph of elongated halite grains showing subgrain-free and subgrain rich halite cores with white overgrowth rims, location is indicated in (b). (e) Reflected light micrograph of grey salt and bent claystone particle and smaller inclusions in-between halite crystals, location is indicated in (b). (f) Crossed polarized transmitted light micrograph showing anhydrite accumulation next to claystone particle and smaller high interference minerals at halite grain boundaries, location identical to (e) is indicated in (b).

## 7.5 Numerical modelling

### 7.5.1 Setup

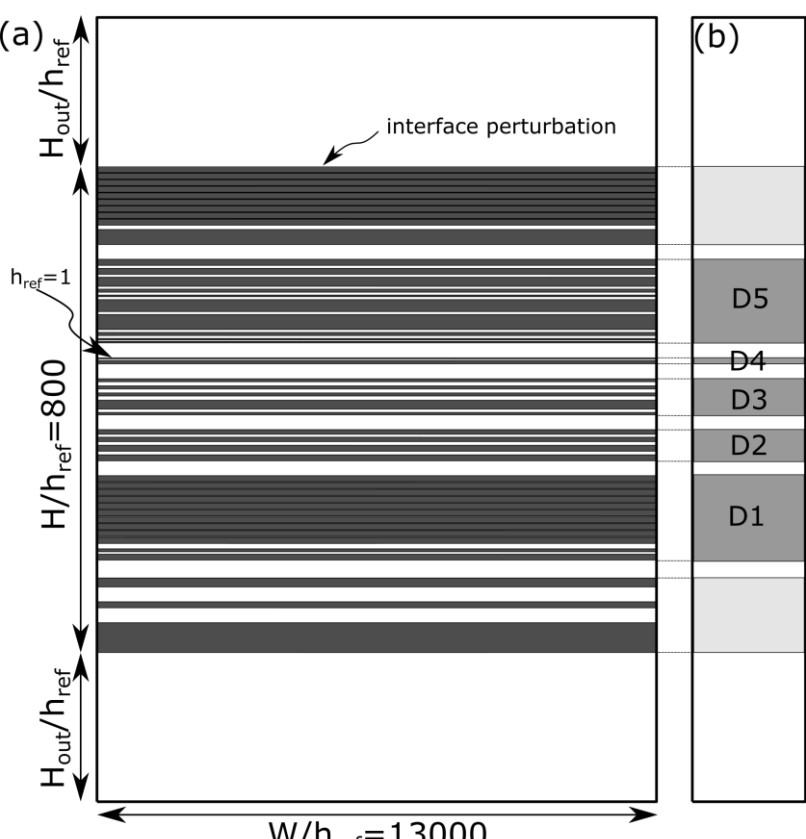

**Figure 12. (a) Model geometry used in the numerical simulations, based on the outcrop shown in Fig. 8. Note that the model has no scale. (b) Domains.**

Multilayer fold structures observed on the south side of the E50 pillar (described in detail in Figure 8) were selected for detailed numerical modelling. We consider a domain comprising a stack of alternating stiff and soft layers with varying thicknesses embedded in the thick soft layers. We constrain the initial thicknesses of the individual layers by calculating their area and divide it by their mean layer arclength. The layer area and arclength values are measured from the digitized photo of the pillar E50 (Fig. 8a). The estimate assumes two-dimensional plane-strain deformation, no volume loss, and initial constant layer thickness. All the length-scale values used in the models are normalized by the thickness of the thinnest layer, $h_{ref}$ (thin layer in the D4 domain).





The computational model width, W/h, is set to 13000. The numerical model consists of a multilayer stack containing 91 layers
with variable viscosities. The thickness of the stack, $H/h_{ref}$, is equal to 800 and the individual layer thickness varies between 1
and 50. On each interface, we initially impose a red noise perturbation. All the layers are treated as incompressible, linear
viscous material and the contacts between the layers are welded.

In all the simulations, the normal components of the velocity vectors are prescribed at the boundaries according to a pure shear
deformation and free-slip boundary conditions are used for all the walls. The model is subjected to up to 90% shortening at
the constant rate of deformation. We use high spatial and temporal resolutions. To avoid significant mesh distortion, the model
geometry is re-meshed at each time step.

In the first set of simulations, we assume a viscosity ratio between dark and white beds of R=8, 10, 15, 20, 30, and 50. The
package is sandwiched between two thick soft layers, whose thickness is equal to $H_{out}/h_{ref}=300$. Moreover, we set the initial
amplitude of perturbation $A_0/h_{ref}=0.15$. In the second set of simulations, we examine the role of the perturbation amplitude,
where we use $A_0/h_{ref}$ equal to 0.045 0.15 and 0.45. Here, we fix R=20 and $H_{out}/h_{ref}= 300$. In the third set of simulations, we
vary the thickness of the outer layers with $H_{out}/h_{ref}=75$, 150, and 300 run the simulations for constant R=20 and $A_0/h_{ref}=0.15$.
The initial parameters applied in the models are presented in Table 1.

We compare the results of the simulations for the sets, when the mean limb dip of the large-scale folds are ca. 45°. However,
this criterion is achieved at different stage of deformation for different models and is listed in Table 1. Additionally, we provide
references to the figures showing the results of the simulations.

**Table 1. Parameters used in the numerical models.**

| Set | R | $A_{pert}/h_{ref}$ | $H_{out}/h_{ref}$ | Shortening [%] | Reference figure |
|-----|-----|-----|-----|-----|-----|
| 1 | 8 | 0.15 | 300 | 85 | Fig. 12a |
| 1 | 10 | 0.15 | 300 | 80 | Fig. 12b |
| 1 | 15 | 0.15 | 300 | 72 | Fig. 12c |
| 1 | 20 | 0.15 | 300 | 65 | Fig. 12d |
| 1 | 30 | 0.15 | 300 | 55 | Fig. 12e |
| 1 | 50 | 0.15 | 300 | 46 | Fig. 12f |
| 2 | 20 | 0.045 | 300 | 67 | Fig. 13a |
| 2 | 20 | 0.15 | 300 | 65 | Fig. 13b |





| 2 | 20 | 0.45 | 300 | 60 | Fig. 13c |
| --- | --- | --- | --- | --- | --- |
| 3 | 20 | 0.15 | 75 | 68 | Fig. 13d |
| 3 | 20 | 0.15 | 150 | 66 | Fig. 13e |
| 3 | 20 | 0.15 | 300 | 65 | Fig. 13f |

**7.5.2    Numerical results**

Figure 13 shows the results of the first set of simulations for different R values. The amount of shortening required to achieve a limb dip of 45° of the large-scale folds decreases with increasing R, e.g., model with R=8 requires 85% of shortening, whereas R=50 is shortened only 43% (see Table 1 for details).

Close spacing between the stiff layers causes that the layers interact with each other. With increasing R, they form a smooth
transition between polyharmonic and harmonic folds. Since very little small-scale fold structures are observed in models with R=30 and 50, we consider that the polyharmonic folds are characteristic for R<30, whereas the harmonic folds occur for R≥30. The influence of R on the large-scale fold structure is best illustrated on the folds geometry of the D1 domain. The increase in R causes the increase in the large-scale fold wavelength and promotes developing of folds with sharper hinges and longer and straighter limbs. Moreover, the thickness of each layer decreases causing that the thickness of the whole D1 domain decreases
with R.

The middle-scale folds tend to develop in the D2 and D3 domains. In Fig. 14a, we compare the shape of the uppermost layer of the D2 domain. The middle-scale folds occur in models with R<30, where the most pronounced structures develop for R=10 and R=15. The middle-scale folds occurring on the limbs of the large-scale folds are the most notable and are generally characterized by an asymmetric shape. For R≥30, the layers have long, nearly straight limbs following the large-scale fold
pattern and develop the 'keel-like' accommodation structure around the hinge zone (Price and Cosgrove, 1990, p.320).

Distinct small-scale folds are represented by thick layer in the D4 domain for R=8 and 10. The thin layer in the this domain locally forms even smaller-scale fold structures. In other models, both layers in the D4 domain fold along middle and/or large-scale folds. For R=50, evident 'keel-like' structures occur in the hinge areas.







**Figure 13. Results of the numerical simulations for viscosity ratio between dark and white layers of: (a) R=8 and shortening 85%, (b) R=10 and shortening 80%, (c) R=15 and shortening 72%, (d) R=20 and shortening 65%, (e) R=30 and shortening 55%, and (f) R=50 and shortening 46%. In the model, the normalized initial amplitude of interface perturbation, $A_{pert}/h_{ref}$, is set to 0.15 and the thickness of the outer, soft layers, $H_{out}/h_{ref}$, is set to 300.**



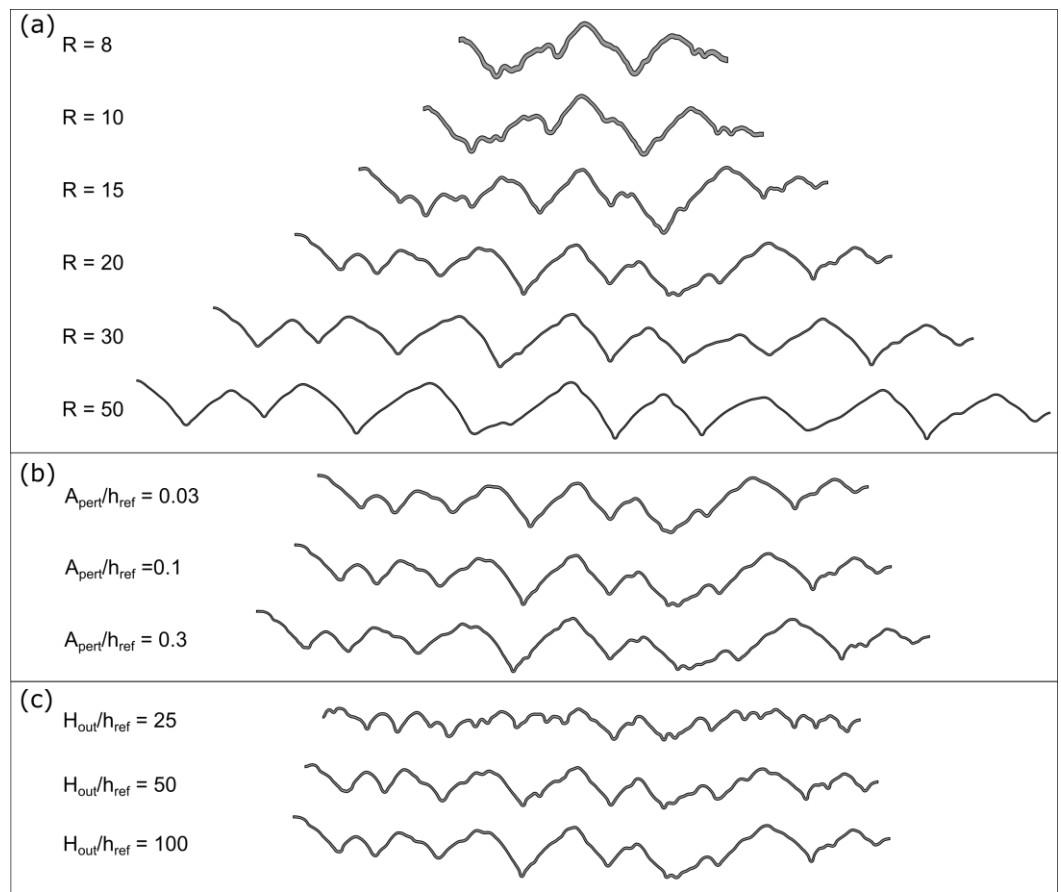


**Figure 14. Fold shape of uppermost layer in the D2 domain for (a) different viscosity ratios, R (set 1), (b) different initial normalized perturbation amplitude, $A_{pert}/h_{ref}$, (set 2), (c) different initial normalized thickness of the outer layers, $H_{out}/h_{ref}$, (set 3).**

In the second set of simulations, we investigate the sensitivity of our results on the amplitude of the initial perturbation (Fig. 15a-c). The large-scale fold shape is generally similar in all the models, however, the decreasing $A_{pert}/h_{ref}$ requires more

shortening to develop into fold limbs of ca. 45°. For $A_{pert}/h_{ref}=0.045$ the results are presented after 67% of shortening, for $A_{pert}/h_{ref}=0.1$ after 65% of shortening, whereas for $A_{pert}/h_{ref}=0.45$ after 60% of shortening.

No significant difference can be observed in the overall shape of the middle-scale folds (Fig. 14b). For larger $A_{pert}/h_{ref}$ the layer interface is rougher. The discrepancy between the models is more pronounced in the case of small-scale structures. For models with larger $A_{pert}/h_{ref}$ more small-scale fold structures can be observed.

The third set of simulations test the influence of the thickness of the bounding layers, $H_{out}/h_{ref}$, (Fig. 15d-f). The folds in models with larger $H_{out}/h_{ref}$ tend to develop folds with larger amplitude and straight limbs. Development of the middle- and smaller-scale structure is more pronounced for models with smaller $H_{out}/h_{ref}$, where the growth of large-scale folds is limited due to confinement (Fig. 14c).







**Figure 15. (a-c) Models showing the role of initial normalized perturbation amplitude for $A_{pert}/h_{ref}$ =0.03, 0.1 and 0.3, respectively. (d-f) Models showing the role of the thickness of the bounding layers for $H_{out}/h_{ref}$ =75, 150, and 300, respectively. In all models, the viscosity ratio between the dark and white layers is set to R=20.**



## 8    Discussion

We combined observations at a wide range of scales of the folded multilayer sequence from the Ocnele Mari salt mine with
numerical simulations to constrain long-term rheological contrast between the layers. The morphology of the folds, at first
look, clearly indicate that the sequence is mechanically stratified and the rheological contrast is associated with the presence
of impurities in different layers. In the sections below, we discuss and interpret our results further, to finally arrive at the
permissible range of viscosity contrast.

### 8.1    Deformation in a salt detachment shear zone

Regional geological arguments, tectonic setting in a fold-and-thrust belt and well data constrain that the salt body is a 400 m
thick sheet conformable to regional layering. The presence of top-to the foreland shear in a salt detachment between clastics
is common in such settings. The salt body is well defined by salt exploration wells (Fig. 1b), and the structure is not a deep-
rooted diapir as shown by reflectance of Vitrinite indicating burial to less than 1.5 km. The faults offsetting the deposit suggest
that deformation slowed down after movement on the detachment.

Structures in the mine (with excellent exposure of clean salt pillars) are in agreement with this interpretation. The layering in
the fold limbs shows evidence for large shear strains and the formation of early isoclinal folds, before the formation of S-
vergent folds with gently curved fold axes, which all agree with deformation in a shear zone which strongly reworked the
original sedimentary layering. Perturbation of the flow field during shearing is inferred to have formed the asymmetry of the
large-scale folds observed in the mine. We suggest that the asymmetry in fold shape in our study are similar to those described
in salt glaciers by Talbot (1979) who attributed it to the presence of irregularities in the basement. Moreover, using numerical
simulations, Schmalholz and Schmid (2012) showed that deformation over a ramp with a slope-break can lead to development
of such fold asymmetry.

### 8.2    Deformation mechanisms and rheology

Optical microscopy of samples representing the hinges and the limbs of the folds studied here was aided Gamma-decoration,
which is a sensitive indicator for small differences in the halite crystal defect structure. In the fold limbs, halite grains show a
strong shape preferred orientation parallel to the foliation, with boudinage of inclusion-rich layers and halite fibres in the
boudin-necks. Overgrowth and truncation structures are common, and subgrain-rich grains are uncommon, suggesting that
solution-precipitation creep was a dominant deformation mechanism. Halite grain size is smaller in the inclusion-rich layers,
suggesting pinning of mobile grain boundaries and enhancement of grain size-sensitive flow. The subgrain-rich grains are
interpreted to have formed in an earlier phase of deformation (the detachment phase?) and are now replaced by fluid-assisted
grain boundary migration.

The shape-preferred orientation is reoriented in the fold hinges to parallelism with the axial plane, indicating that the grain
fabric is related to folding. Microstructures representing this deformation show evidence for solution-precipitation creep





indicated by dislocation-free grains with truncation and overgrowth structures. This we interpret for evidence for solution
precipitation creep, which is indicative for linear (Newtonian) viscous rheology, supporting the choice of rheology in our
numerical models. The darker layers are inferred to contain more impurities and have higher viscosities. However, this
microstructure does not provide information on deviatoric stress (solution-precipitation creep does not form subgrains), so we
have no way to test if the layers inferred to have had higher viscosities actually deformed under higher deviatoric stress as
expected, we only note that deviatoric stress was very low (less than 1 MPa). We also note that in the limbs of the folds where
the subgrains may represent the deformation in the higher stress detachment phase, the similar deviatoric stresses in the
different layers (Supplement) may reflect the homogeneous state of stress (but not shear strain) expected to be present in simple
shear.

### 8.3 Folds in the Ocnele Mari salt mine

In the Ocnele Mari salt mine, polyharmonic fold structures occur on up to four scales. As shown by Ramberg (1962),
polyharmonic folds can develop in the multilayer sequence, where significant variation of the more competent layer
thicknesses is observed. However, in the Ocnele Mari salt mine, the salt is finely layered and the variation of the layer
thicknesses ranges between centimetres to tens of centimetres with a maximum of ca. 30 cm. In this study, we show, the
development of polyharmonic folding is not attributed to the variation of the individual layers but to the arrangement of the
fine layers in the multilayer stack. Close spacing of package of thin layers causes that this package behaves effectively as thick
single layer triggering structure development on larger scale. Consequently, the assemblage of layers and packages of layers
can result in the deformation on various of scales.

Generally, the dark layers develop folds with rounded hinges and they are, thus, considered to have higher viscosity as compare
to the bright layers (Figs. 4c,d 5a-d). We suspect that mechanical properties of the dark layers can slightly differ between each
other as they show different shade of grey colour, which is probably related to the variable amount of impurity in the layer.
The effective viscosity of the multilayer packages forming an effective single layer would depend on the characteristic of the
individual dark layer, number of these layers, and their thickness.

However, some of the very thin grey layers form folds with sharp hinges and wavy limbs following the shapes of the
neighbouring layers (Figs. 4c and 5d). These structures are more characteristic for passive layers. We infer that the layers are
not mechanically homogeneous, due to their relatively small thickness compared to the size of impurities and salt grains.
Similar observations have been reported for the case of boudinaged structures by von Hagke et al. (2018).

### 8.4 Numerical model design

The most critical factor in designing our multilayer modelling was the estimation of the initial layer thicknesses. We constrain
the parameters based on the relative area of the layers that we observe in the outcrop. The estimate assumes plane strain
deformation, no-volume change in the deformed rock mass and initially constant lateral layer thicknesses. Further uncertainties
in the definition of the layer thickness are related to the digitization error and definition of the layer thickness. The problems





are greater for the case of very smaller layers or layers with diffuse boundaries. We expect that the error related to the layer thickness estimation does not significantly affect our numerical simulations. On one hand, minor changes are expected to occur if the wrongly estimated layer is a part of the multilayer package. On the other hand, more pronounced changes could occur for the isolated layers, however, this would not affect the overall structure development.

The other problems related to the design of the numerical model include defining: boundary condition and initial perturbation type and amplitude. We incorporate these factors in the numerical analysis and investigate their influence on the developing structures (see below).

## 8.5 Viscosity ratio estimates

We attempt to investigate the viscosity ratio in the folded multilayer sequence using two approaches: a) based on the single 605 layer fold shape analysis and b) testing the occurrence of polyharmonic folds.

### 8.5.1 Single layer fold shape analysis

We carried out the analysis of the fold shape that develops in the lower layer of the D4 domain, which is embedded by white layers with relatively considerable thickness. The upper layer in the domain is very thin and generally illustrates a passive layer behaviour.

We choose a part of the layer with small thickness variation to reduce unnecessary complexity of the analysis and increase reliability of the solution. Two independent methods developed by Fletcher and Sherwin (1978) and Schmalholz and Podladchikov (2001) for estimating viscosity ratio of the folded single layer indicate values between 10 and 25.

### 8.5.2 Polyharmonic folding

Development of polyharmonic folds requires a range of geometrical and mechanical conditions to be satisfied. These 615 constraints provide an excellent opportunity to investigate the evolution of our multilayers. We carefully digitized the outcrop in the pillar E50, where the three orders of folds can be observed, and designed a model for detailed numerical analysis. We simplified the number of geometrical parameters by constraining the relative layer thicknesses. For varying viscosity ratios between the layers, we investigate development of polyharmonic structures.

In the numerical simulations of multilayers, the polyharmonic structures appear for models with R<30. An overall good 620 correspondence of the computed structures and field observations is obtained for R=10 and R=15. In both cases, we can observe developing folds over three distinct orders with morphologies that can be matched with observations.

In the models with R=8, we observe some discrepancy in the observed and numerically generated large-scale folds. The numerical folds are characterized by much smaller amplitude and length of the limbs. The shape of the structures is much more concentric-like when compared to the field examples, and these models require very large shortenings which are unlikely in 625 our natural examples.



Numerical models with R=30 and R=50, generate large-scale folds with sharp hinges. The multilayer sequence folds harmonically and the development of the smaller-scale structures is suppressed. Moreover, layers in the domains D2 and D3 tend to develop 'keel-like' accommodation structure around the hinge zone of the larger folds. These structures are not observed in the Ocnele Mari salt mine.

With increasing viscosity ratio, the amount of shortening needed for the large-scale structure to reach 45 limb dip decreases. Our field observations allow us to constrain the minimum shortening, which is 35-45%. These values are still below the analysed model of R=50, which is shortened by 46%. Nevertheless, this constraint shows that viscosity ratios were less than R=50.

In the model, we used only two viscosities, which correspond to white and dark layers in the field. Thus, in our estimates, we
refer to their effective viscosity ratio and we do not take into account more than one value of viscosity in the dark layers or lateral changes in their composition. As illustrated in Fig. 4d, yellow-brown layers are probably more competent than black layers.

Since the selected example of the sequence in pillar E50 is characterized by a rather simple configuration of the layers with generally sharp layer interfaces, we argue that this simplification is acceptable. In future work we aim to analyse a core through
the different layers, to characterise the impurities leading to the different shades of grey. Note, that the detailed relation between the impurity content and different shades of grey and their influence on the effective viscosity of salt is not known in detail and is beyond the scope of this paper.

### 8.5.3 Number of layers in the multilayer sequence and confining medium

We analyse structures developing in a selected part of a much larger multilayer sequence. In the numerical model, we simplify
this far field by introducing thick upper and lower weak layers. Above and below the analysed sequence (D1-D5), we include additional layers to decrease the boundary effect.

As pointed out by Biot (1961), dominant wavelength increases by a factor of $m^{1/3}$ for a multilayer composed of m-layers. The relation is valid only for the case of simple multilayer stack composed of alternating stiff and soft layers with equal thickness. Increasing the number of layers in the analysed sequence would result in a larger dominant wavelength of the large-scale folds.
This can affect the development of the middle- and small-scale folding. On the other hand, including a thick weak layer around the multilayer sequence promotes the development of large-scale folds (Fig. 15d-f). Significant change in the fold shape evolution is found only for the close proximity of the rigid boundaries, which is not the case we observe in the field. Consequently, we argue that the simplification used in the numerical model does not change our findings.

### 8.5.4 Initial perturbation amplitude

Initial perturbation of the layer interface is the parameter that is difficult to constrain. A common practice in numerical simulation of folding is using red noise perturbation with the initial amplitude normalized by the layer thickness. However, for the multilayer sequence composed of variable layer thickness this assumption cannot be used. Instead, we use equal





perturbation amplitude for all the layers. Since the presence of impurities in the layers are probably the main source of the natural perturbation, we consider this assumption to be reasonable, following Frehner and Schmalholz (2006).

Increasing initial perturbation amplitude promotes development of large amplitude folds, which effectively reduces the amount of shortening needed for large-scale folds to reach limb dip of ca. 45°. Moreover, a large initial amplitude reduces the wavelength selection process. Consequently, the layer interface is rougher and the regularity of the developing fold shapes is reduced. In our simulations, we see more pronounced small-scale folds developing in the models with initially higher $A_{pert}/h_{ref}$.

## 9 Conclusions

• The field observation and numerical models clearly show that the Ocnele Mari impure rocksalt is mechanically heterogeneous and anisotropic during deformation at low differential stress, below 1 MPa.

• For a relatively small fraction of impurities (ca. 10%), viscosity ratio up to 20 can form in fine-layered rocksalt. We propose that such contrasts are common, giving layered rocksalt an anisotropic viscosity at a wide range of scales, in addition to the effects of layers of anhydrite, clay, and K-Mg salts.

• Microstructural analysis shows evidence of dissolution-precipitation processes, supporting our assumption of Newtonian viscous creep under these conditions.

• Development of polyharmonic folds requires a range of geometrical and mechanical conditions to be satisfied, providing a tool for constraining viscosity ratios in ductile deformation.

## 10 Author contribution

MA designed and carry out the numerical analysis. DMT provided the field data and the photos. JB performed the microstructural analysis. JLU designed the project. All the co-authors contributed to writing the manuscript.

## 11 Competing interests

The authors declare that they have no conflict of interest

## 12 Acknowledgements

MA was supported by Polish Geological Institute - National Research Institute research project no. 62.9012.2054.00.0. DMT acknowledges funding from Babeş-Bolyai University, through GTC grant no. 35275/18.11.2020.

JLU acknowledges the hospitality and inspiring discussions of Stefan Schmalholz and his Group during a sabbatical at University of Lausanne, where the first ideas of this study were developed. This sabbatical was supported by a Theodore von





Kármán Fellowship of RWTH Aachen University. We thank Lina Gotzen for providing Fig. 6 from her MSc study at RWTH
Aachen University.

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
