# Peer review of "Rheological stratification in impure rocksalt during long-term creep: morphology, microstructure and numerical models of multilayer folds in the Ocnele Mari salt mine, Romania"

_Solid Earth, 2021_

## Referee Comment (RC2)

[referee-annotated manuscript omitted]

---

## Author Comment (AC1)

**Answers to Review #1**

**Specific comments:**

*Q1: Line 307: Please indicate the diffuse boundaries in Fig. 4e and 5b. You don't refer to Fig. 4f and Fig 5d.*

**Response:** Diffused boundaries are best observed in Fig. 5b and d and we indicated them using the arrows. We fixed all the references to figures.

*Q2: Line 392-393: I would put this in the method*

**Response:** Done. The sentence was moved to the methods section of the manuscript.

*Q3: Line 406: are these truncations indicated by one of the white arrows? If not, can you indicate them in the figure.*

**Response:** Done. We indicated it in the figure.

*Q4: Line 411: Please add a figure reference for "subgrain boundaries appear white in blue halite cores".*

**Response:** Done. The reference was added.

*Q5: Line 420: where are the fluid inclusions in Fig. 11f?*

**Response:** This was wrongly referenced in the text and corrected to "11e".

*Q6: Line 422: I don't see fibrous crystals in Fig 11e.*

**Response:** This was wrongly referenced in the text and corrected to "11c".

*Q7: Line 424: Please explain why you refer to Krabbendam et al., 2003 here.*

**Response:** This paper is a good example of a recrystallized texture where grain size is strongly affected by second phases.

*Q8: Line 557: subgrain rich grains are uncommon but in line 437 you say abundant presence of subgrains, also visible in Fig. 10d and 11d?*

**Response:** For clarification, we have added: "Further, we suggest that most subgrains visible through gamma decoration but not visible in the etched surfaces are paleosubgrains, which are not present anymore although this cannot be proven by the presented data."

*Q9: Line 632: Why does this constraint show that the viscosity ratios were less than R=50?*

**Response:** We corrected the sentence "This constraint shows that viscosity ratios could be slightly higher than R=50".

**Technical corrections**

*Q10: Line 49: delete one MPa*

**Response:** Fixed.

*Q11: Line 91: The shape*
  **Response:** Fixed.

*Q12: Line 304: remove folds*
  **Response:** Done.

*Q13: Line 307: a locally diffuse boundary or locally diffuse boundaries*
  **Response:** We wrote "locally diffuse *boundaries*".

*Q14: Fig. 10a) what are the green lines in the figure? c & d: what do the white arrows point to? There is no (f) at the beginning of the description of (f).*
  **Response:** We have added in the caption of Fig. 10:
  a) "Green lines indicate folded millimetre-scale inclusion rich halite layers."
  c) "Arrows point to grain boundary of halite grain that is overgrowing fluid inclusion bands of neighbouring grain.
  d) "Arrows indicate blue and white subgrainboundaries."
  f) fixed.

*Q15: Table 1 – Change Fig. 12 to Fig 13, and Fig. 13 to Fig. 15 Perhaps good to somewhere indicate that Fig. 13d, 15b and 15f are the same.*
  **Response:** Thanks for noticing. We fixed it.

*Q16: Line 499: Moreover, the thickness of each layer decreases causing the thickness of the whole D1 domain to decrease*
  **Response:** Done.

*Q17: Line 506: represented by the thick layer*
  **Response:** Done.

*Q18: Line 506: Please refer to Fig. 13a &b for the distinct small folds.*
  **Response:** Fixed.

*Q19: Line 506: The thin layer in this domain*
  **Response:** Done.

*Q20: Figure 15 caption: $A_{pert}/h_{ref}$ 0.03 should be 0.045, 0.1 should be 0.15 and 0.3 should be 0.45.*
  **Response:** Thanks for noticing. We fixed it and also corrected the same mistake in Fig. 14.

*Q21: Line 577: In this study, we show that*

**Response:** Done.

*Q22: Line 579: effectively as a thick*

    **Response:** Done.

*Q23: Line 607: Refer here to Fig. 9*

    **Response:** Done.

---

## Author Comment (AC2)

**Answers to Review #2**

Please note that the comments numbering continues from the Answer to Review #1.

**General comments:**

*Q24: (...) My general impression is that there are two themes in the manuscript that are not well linked - the microstructural analysis of the salt rocks and the numerical modeling of the multilayer folding.*

*(...)*

*Altogether, the microstructural part of the manuscript fails to deliver the relevant information from the detailed description and quantification of the microstructures. In the results part and in the discussion, the outcomes shrink to a couple of weakly based statements that aim to bridge the microstructural part to the numerical modeling of the multilayer folding:*

*1) the dominant deformation mechanism is grain size-sensitive solution-precipitation creep (which justifies for the Newtonian viscous num. model),*

*2) the solid-inclusion rich layers (dark layers) are more susceptible to the grain size-sensitive flow,*

*3) the deviatoric stresses were very low.*

*(...)*

*I suggest for the reorganization of the manuscript, sections 1 and 4 can be merged and shortened, the microstructural part needs to be complemented and related discussion improved.*

**Response:** The goal of the paper is to gain an insight into the long-term rheological behaviour of the impure rocksalt in the Ocnele Mari salt mine. We aimed to use a combined approach of field observations, microstructural analysis, and numerical modelling to obtain the most comprehensive results. The microstructural analysis is not the main theme of the paper. Instead, we use it to infer the dominant deformation mechanism and determine either Newtonian or non-Newtonian rheological behaviour of rocks. This information is further used to constrain the parameters employed in the numerical models.

We modified the paper to motivate the use of the microstructural analysis and to provide the relevant information that supports the conclusion that our data strongly suggest the Newtonian rheology of the impure rocksalt.

Moreover, we disagree with the suggestion about merging Sections 1 and 4. Analysis of the multilayer buckle folds is the main focus of our paper and in the section 4, we presented the key information that is further used in the study. Thus, for clarity and transparency reasons, we prefer to keep it as a separate section and do not simplify the text.

*Q25: The introduction to the microstructures and rheology of rocksalt is long and presents the general overview of the present knowledge about the rheology of rocksalt and methods that allow estimation of rocksalt's long-term flow properties. This introduction also mentions the viscosity differences in evaporitic sequences, in terms of enclosed layers of different compositions (e.g. low-viscosity, potassium salts, or anhydrite). However,*

*the introduction neglects the rheological effect of solid second phases (impurities) in halite-dominated rock salt. This topic was studied for example by Závada et al. (2015). This latter study is particularly mentioned, because Závada et al., (2015) came to completely different conclusions than the authors of the present study. Závada et al., (2015)* *explained that rocksalt, which is rich in dispersed solid second phase particles, can become weaker (in terms of viscosity) than the pure salts* *because the pressure solution-precipitation creep rates are much faster on boundaries of the solid phases (impurities) with halite crystals than on halite-halite boundaries. Therefore, in this perspective, how is it possible that the fine-grained material of the darker layers in the folded sequence of the Ocnele Mari salt mine is stronger in deformation than the white (more pure) layers?*

**Response:** We modified and shortened the section discussing the viscosity of different rocks in the evaporitic sequences (anhydrite and bittern salt). Moreover, we extended the paragraph to include information about the role of the second phase minerals on the effective viscosity of rocksalt. We mentioned that some studies, including the paper of Závada et al. (2015), report the weakening effect of the solid second phase particle, whereas other studies, such as Jordan (1987), indicate their strengthening effect.

Our field observations clearly point that the dark (impure) layers are more viscous than the white (pure) layers. The in-depth understanding of the processes that result in the strengthening of the impure salt layers is, however, a subject for a separate study and requires detail microstructural analysis on a much larger number of samples.

*Q26: First, you clearly see that both dislocation creep and solution-precipitation (SP) creep were active at the same time (see Fig. 11d, where white parts of grains are full of subgrains and locally mark the "necked" domains of the long grains) - so how clearly dominant was the SP creep? Can we estimate the relative contribution of dislocation creep and SP creep?*

**Response:** We suggest that most subgrains (indicative for dislocation creep) visible through gamma decoration but not visible in the etched surfaces are paleosubgrains, which are not present anymore, although, this cannot be proven by the presented data.

The observations presented here on paleosubgrains have been made for the first time, and these could be the start of new investigations.

For clarification, we have added: "Further, we suggest that most subgrains visible through gamma decoration but not visible in the etched surfaces are paleosubgrains, which are not present anymore although this cannot be proven by the presented data."

*Q27: Second, small grain size and small differential stress supporting the SP creep should be linked to lower viscosity in contrast to the coarser-grained (white salt) equivalent. So - why is the dark salt layers more viscous than the white layers???*

**Response:** Based on field and fold shape observations, and we infer that the dark layers have higher viscosities. However, to better quantify the role of the amount of impurities, their shape, and distribution on the effective

properties of the rocksalt, more samples and analysis are needed. Detailed microstructural analysis is beyond the scope of the article.

**Q28:** *Third - the deviatoric stress estimates span from 0.4MPa to 4Mpa in the same thin section - so the low deviatoric stresses (below 1MPa) are simply unfounded and the high-stress estimates can not be discarded simply by attributing it to the older (detachment shear) deformation event.*

**Response:** We agree with the reviewer that the unusually wide range of subgrain sizes (not expected based on the intrinsic anisotropy of Halite) suggests that subgrains were formed in multiple episodes of deformation and recrystallization. However, here we take a practical approach to average the data, because we do not have the larger number of samples required for investigating this in detail.

**Q29:** *The text explains how grey, dark, white, and yellow-brown layers are folded together, however, only white and dark layers are visible in the black and white images of the fold patterns (Figs. 4 and 5).*

**Response:** We changed the figures, so the grey, white, and yellow-brown layers are more visible.

**Q30:** *The total amount of insoluble second solid phases is not indicated anywhere in the manuscript (only the relative contents of phases in the residuum and in the Conclusions that is somewhat below 10 wt.%).*

**Response:** In this study, we chose not to quantify the amount of impurities due to the limited amount of specimens. For a comprehensive and representative impurity analysis, more samples and further analysis are required.

In the discussion section 8.2, we added: "However, to better quantify the amount of impurities more samples and analysis are needed."

**Q31:** *Since two phases of deformation are recorded in the folded sequence, it is difficult to judge on the meaning of the subgrain size piezometry. Nevertheless, since the results of the piezometry is not used for implementation of the boundary conditions of the numerical model, it could be simply removed.*

**Response:** See comments Q28 and Q78.

Moreover, it is interesting to note that subgrains have variable mean subgrainsizes and that we observe no significant difference between grains in the white and the dark layer. We have added "Subgrain size piezometry (Schléder and Urai, 2005) of RO-OM-1 indicates variable mean subgrain sizes for individually measured halite grains with no significant difference between the dark and the white salt (S.1.2 in Supplement)".

This also strengthens the interpretation of dominant deformation by solution-precipitation processes as stated in the discussion section. "We note that pressure solution microstructure does not provide information on deviatoric stress (solution-precipitation creep does not form subgrains), so we have no way to test if the layers inferred to have had higher viscosities actually deformed under higher deviatoric stress as expected, we only note that deviatoric stress was very low (less than 1 MPa)."

***Q32:*** *The microstructural part should try to address and discuss the microscale effects of solid second phases in the developing folds and rates of material exchange between the white and dark layers. For example - I see that in Fig. 11e, a single crystal of halite is able to distort a thin layer of dark salt (formed by flakes of clay and halite) - therefore, locally, it is clear that halite is mechanically stronger than the dark particle bearing salt. This is to say that the microscale effects need to be properly discussed, even though it seems (from macroscopic fold geometry) that the dark layers are stronger.*

**Response:** We agree with the reviewer that the comprehensive microstructural analysis would be very inspiring, however, it requires more systematic study on a much larger number of samples. Such analysis is beyond the scope of this paper.

Regarding the note about the halite mineral, we have added in the discussion section: "We consider the dark salt to be more competent than the white salt based on field and fold shape observations. We note that the dark salt itself consisting of claystone particles and halite grains can have internal competency contrasts (Fig. 11e), which are not resolved in our numerical models."

***Q33:*** *The numerical modeling part is presented in a somewhat more coherent way and requires another lengthy introduction (section 4 - presented after "geological setting"), which can be reduced in places (see the annotated pdf for suggestions).*

**Response:** See comment Q24.

***Q34:*** *It seems that the results of the numerical modeling are supported by the validity of the thin single layer fold estimates using FGT toolbox (Figure 9) and the complex multilayer modeling. By presenting viscosity estimates by analysis of fold geometry of the two different models (Fletcher and Sherwin (1978) and Schalholz and Podladchikov (2001)), one should also explain, how are they defined.*

**Response:** We extended the description of the parameters used in both methods and included in the diagrams in Fig. 9. For this reason, we also extended Section 4 to introduce the concept of preferred wavelength.

However, since the analysis of the single layer fold shape is not the main subject of our paper, we provide a reference to the work of Hudleston and Treagus (2010), which shortly presents and discusses these two methods.

***Q35:*** *Could the numerical modeling part try to provide clues for how would the microscale material transfers (local velocities of solution-precipitation controlled by grain size and solid inclusions) affect the geometry of folds and the viscosities?*

**Response:** The codes that we used in the project are not designed for such an analysis. In order to determine the relation between the deformation in microscale and the effective behaviour of the material, the numerical model should include the microstructural processes. Our codes use only the up-scaled material properties.

***Q36:*** *Since the viscosity contrast of 10-15 is quite high, would it be possible to estimate the bulk viscosity of the multilayer in contrast to pure salt on the basis of the modeling? This would be potentially interesting for the*

*salt tectonics community, where people are modeling the development of salt structures (e.g. by numerical or analog modeling).*

**Response:** Thank you for mentioning this. The presence of the mechanical layering introduces anisotropy in the system, so the "bulk" viscosity would depend on the orientation of the layers. To note that, we included an additional section in the paper to discuss this subject (Section 7.6). Moreover, we speculate that the evaporite sequence might be commonly anisotropic.

**Specific comments:**

**Q37:** *Lines 12-16: This text sounds more like an intro rather than the abstract...*

**Response:** We modified and shortened the paragraph. We wrote: "At laboratory time scales, rocksalt samples with different composition and microstructure show a variance in steady state creep rates, but it is not known if and how this variance is manifested at low strain rates and corresponding deviatoric stresses. Here, we aim to quantify this from the analysis of multilayer folds that developed in rocksalt over geological time scale in the Ocnele Mari salt mine in Romania. (…)"

**Q38:** *Line 21: I would use "shapes", as the 3D 'morphology' was not really constrained in this MS.*

**Response:** We modified the text: "Fold patterns at various scales…"

**Q39:** *Lines 25-27: This sentence should be split in two.*

*How can you indicate that the salt was Newtonian viscous, if both - dynamic recrystallization (driven by migration of dislocation and by definition power-law flow) and solution-precipitation creep (Newtonian) were clearly active together?*

**Response:** Done. We split the sentence in two. We wrote: "Microstructures indicate dislocation creep, together with extensive fluid-assisted recrystallization and strong evidence for solution-precipitation creep. This provides support indicative for linear (Newtonian) viscous rheology as a dominating deformation mechanism during the folding."

Regarding the comment about the deformation mechanism, please see Q78.

**Q40:** *Lines 36-37: how pure? The amount of insoluble second phases is not specified anywhere in the MS (only relative content of different phases in the residue)*

**Response:** See comment Q30.

**Q41:** *Lines 61-62: Instead of specifying all studies that employed microstructural analysis, accentuate those that are most relevant for your study dealing with impure salt.*

**Response:** We thank the reviewer for this comment. We extended the paragraph below and included the references that discuss the role of the second phase on the effective mechanical properties of the rocksalt.

Regarding the references provided here, we believe that they are all relevant for the study.

*Q42: Lines 67-68: Erosion of salt or the overburden of course leads to decreased load. I am not convinced that the salt flow on Mt. Sedom (Weinberger et al., 2006) or Hormuz island (Mukherjee et al., 2010) is driven by removal of overburden. It is rather the result of buoyancy of salt caused by differential loading of the overburden.*

**Response:** We wrote: "surface displacement field in areas of active salt tectonics and, in the areas, where erosion led to a change of overburden load".

*Q43: Line 65: Change "ice" into "salt"*

**Response:** See comment Q42.

*Q44: Line 68: You should definitely cite Bruthans et al. (2006) who first argued about the rheology of rock salt from the extrusion profiles.*

**Response:** Done.

*Q45: Line 69: Cite here also the paper of Zurab Chemia*
*https://www.sciencedirect.com/science/article/pii/S0191814108001090*

**Response:** Done.

*Q46: Lines 74-90: I do not see a direct link of this text to the objectives, result of the manuscript. The present paper deals with impurity - second solid phase bearing halite aggregates. The different composition of the salts (bittern salts) or a different lithology (anhydrite layers) are therefore irrelevant. Fold patterns of bittern salt or anhydrite layers in layered evaporite sequences are not compared to the fold patterns in the Ocnele Mari salt mine, or? So, why is it important?*

**Response:** The main goal of this section is to highlight that evaporite succession is mechanically stratified. The mechanical stratification can be attributed to (i) intercalation of different rock types (which is commonly discussed in the literature) or (ii) different content of impurities in rocksalt layer (which is rarely discussed in the literature and it is the subject of the paper). We also present the possible range of competency contrasts, which we further refer to in Section 7.6.

We shortened and modified the paragraph to better show the importance of the presented information.

*Q47: Line 85: what do you mean by "rheological contrast"?*
*Is rheology a parameter expressed by a quantity? No - it describes mechanical properties of the rock as a whole and is expressed by a set of parameters e.g. within a flow law. The rheology of rock (salt) can be deformed by different mechanisms over time and the present microstructure represents the frozen-in state of*

*the deformation. Therefore, it should be "(effective) viscosity contrast" between the layers, that we try to constrain.*

**Response:** We fixed it and used "competency contrast" in the paper.

*Q48: Line 101: ...not sure if this is supported by the present data.*

**Response:** We modified the sentence indicating that our data strongly suggests the Newtonian rheology of salt. Please, see also comment Q78.

*Q49: Line 107: there are two units shown by blue color in the legend, differentiate them*

**Response:** We fixed it.

*Q50: Line 109: the rectangle indicates location of the "local geological profile"... This profile is not shown in the manuscript.*

**Response:** The "regional geological profile" is shown in Fig. 1a, whereas the "local geological profile" is shown in Fig. 1b. In Fig. 1a, red rectangle indicates position of the local geological profile. In Fig. 1b, red rectangle shows location of the mine. To avoid confusion, we modified the colour of the rectangle in the subfigure Fig. 1b, added a text, and changed the caption accordingly.

*Q51: Line 144: This is another introductory text and should be simplified and presented in the introduction of the manuscript.*

**Response:** Please, see comment Q24.

*Q52: Line 182: this paragraph belongs to discussion*

**Response:** As the paragraph presents the current state of the knowledge about the polyharmonic folds, we prefer to leave it in this section.

*Q53: Line 191: is this section relevant, important? Maybe one brief sentence that justifies the numerical modeling setup is enough.*

**Response:** We shortened and merged this paragraph with the section above (Section 4).

*Q54: Line 229: the diffence between both methods should be explained and discussed, when the results of both are presented next to each other in Figure 9*

**Response:** See comment Q34.

*Q55: Line 230: you should explain either here or at the end of Introduction what is the purpose of the microstructural analysis...*

**Response:** See comment Q24.

*Did you want to find evidence for the mechanical properties of the folded layers - compare the microstructures in white and dark layers? What did you expect, how they might differ? Why are the dark layers apparently so much more viscous than the white salt layers?*

**Response:** See comment Q31 and Q32.

*Q56: Line 259: draw a rectangle to show, which pillar was documented in detail (Figure 3).*

**Response:** Done. We modified the figure and changed the caption.

*Q57: Lines 292-293: the colors (yellow-brown) can not be identified on black and white images of Figure 4... This must be fixed - either provide color photographs or add a sketch with indicated color variations. There are white, grey, black, yellow brown layers in the text. In figures I only see dark and white...!*

**Response:** Done. We modified Figs. 4 and 5.

*Q58: Line 295: the small scale folds in Fig. 4c can not be identified - provide a close up view in inset*

**Response:** We zoomed in on the structure in the figure, so the small-scale folds are more visible.

*Q59: Line 304: modify the sentence*

**Response:** We corrected the sentence: "we observe variable geometry of the fold shapes developing on different scales."

*Q60: Line 332: indicate by rectangle location of figure 9a*

**Response:** Done.

*Q61: Line 267: ok, both calculation methods show same result. Is this significant - does it add up on the validity of your calculations? How are both methods different. This deserves more detailed explanation (...and discussion).*

**Response:** See comment Q34.

*Q62: Line 380 (Fig. 10a): "anhydrite" tag in subfigure (a) is not visible*

**Response:** Done. We modified the figure.

*Q63: Lines 380 (Fig. 10d): what do the arrows in (d) indicate? Specify!*
*I am not convinced that they represent "growth rims"!*

**Response:** Fixed. The arrows indicate blue and white subgrain boundaries. We modified the caption to include this information.

***Q64:*** *Lines 410-415: I do not understand what is the purpose of the subgrain size quantification in the paper. Methdological uncertainties in identifying the subgrains are irrelevant.*

**Response:** We interpret the subgrains to constrain the maximum past differential stress in our samples. Therefore, the deformation in our folds took place at lower differential stress. In addition, we suggest that most subgrains visible through gamma decoration but not visible in the etched surfaces are paleosubgrains, which are not present anymore although this cannot be proven by the presented data.

The observations presented here on paleosubgrains have been made for the first time and could be the base for new investigations.

***Q65:*** *Line 417: indicate grain size first...*

**Response:** We prefer to keep the order of observations.

***Q66:*** *Line 419: where is the anhydrite in Fig. 10b?*

**Response:** We have indicated an anhydrite clast in sub-figure 10b. Please note that the large anhydrite clast from Fig. 10a is not part of sample RO-OM-01.

***Q67:*** *Line 421: I do not understand. How are anhydrite grains "intergrown" with halite matrix?*

**Response:** For clarification, we have added: "(…), which is best seen in Fig. 11f where large anhydrite needles penetrate the adjacent halite grain."

***Q68:*** *Line 423: Insert "second phase"*

**Response:** Adapted as suggested.

***Q69:*** *Line 427: just specify the thickness in mm or micrometers /e.g. in contrast to other layers in the section, avoid using adjectives expressing the relative quantities*

**Response:** Adapted as suggested. We wrote: "We observe two approximately 1 millimetre thick "layers" of impure salt outside the fold (…)".

***Q70:*** *Line 429: What is the total amount of the impurities in the layers? This is nowhere indicated, although it is mentioned that it is relatively low (below 10wt. %) in the Conclusions.*

**Response:** See the comment Q30.

***Q71:*** *Lines 446-447: what is the point of averaging the differential stress estimates that span from 4MPa to 0.4MPa, when you argue that there were two events that may be responsible for deformation of the investigated salt...*

*The high stress at 4Mpa might be*

**Response:** See comment Q28.

*Q72: Line 448 (Fig. 11a): what is the grey/dark salt here?*

**Response:** Adapted in the figure.

*Q73: Line 455: ok - I see a dark layer, which is clearly bent, offset by the halite porphyroclast,*
*now - what is stronger: halite porphyroclast (white layer) or dark layer?*

**Response:** See comment Q32.

*Q74: Line 464: Are there any clues from the result of the microstructural analysis that might be significant for*
*construction of the numerical models? Also the simplification of the geometry and the rheology of the natural*
*sequence formed by different layers of variable impurity content (color) should be briefly explained.*

**Response:** See comment Q35.

*Q75: Line 509 (Fig. 13): include the different parameters in the insets of the subfigures (the R parameter and the*
*amount of shortening) - this makes it easier to compare the results (same for Fig.15)*

**Response:** We fixed the figures.

*Q76: Line 529 (Fig. 15): Split the figure into two figures - as the top half and bottom half show effects of variation*
*of different parameters!*

**Response:** Done. We changed the figures.

*Q77: Line 549: Asymmetric folds can be produced anywhere where the flow lines or plane of maximum finite shear*
*is oblique to the sedimentary layering. You do not need obstacles to form them.*

**Response:** We disagree with the reviewer on this point. Various studies show that orientation of the layer oblique
to the shortening direction does not lead to development of the asymmetric folds (e.g., Ghosh 1968; Schmalholz
and Schmid, 2012; Llorens et al., 2013). In the case of linear viscous rheology of the rocks, the shape of the folds
depends mainly on the bulk layer-parallel shortening. We discuss it in the paper in Section 4.

The fact that we observe in the large-scale asymmetric folds is a subject of our discussion, where we speculate
that it can be related to the presence of irregularities in the basement. The formation of asymmetric folds over
irregularities in the basement was described by Talbot (1979) and modelled numerically by Schmalholz and
Schmid (2012).

*Q78: Line 553: This whole section is simply inadequate to the extent of microstructural analysis and space for its*
*results as they are presented in the manuscript. The inferences are also incorrect or not suitably justified.*
*This section claims that:*
*1) the dominant deformation mechanism is grain size-sensitive solution-precipitation creep,*
*2) the solid-inclusion rich layers (dark layers) are more susceptible to the grain size-sensitive flow*

*3) the deviatoric stresses were very low???*

*First, you clearly see that both dislocation creep and solution-precipitation (SP) creep were active at the same time.*

*Second, small grain-size and small differential stresses supporting the SP creep should be linked to lower viscosity in contrast to coarser grained (white salt) equivalent. So - why is the dark salt layers more viscous than the white layers???*

*Third - the deviatoric stresses estimates span from 0.4MPa to 4Mpa in the same thin-section - so the low deviatoric stresses are simply unfounded.*

**Response:** We thank the reviewer for his question, which has made clear to us that we need to clarify better our interpretation that (i) the folding occurred in a later, lower differential stress phase, after the higher stress, shearing phase; (ii) in this second phase, we interpret the deformation mechanisms to be dynamic recrystallization and solution-precipitation creep, while the grains with small subgrains represent the earlier phase; (iii) dominant solution-precipitation creep in the folding suggests Newtonian creep, providing support for the rheology chosen for the numerical modelling.

We have added some words and slightly rewrote the section to clarify it better.

*Q79: Lines 566-569: This sentence does not make sense:*

*you say there is no way of finding if the differential stress was high ("as expected" - expected by who? - authors of the MS?), but you claim that the deviatoric stress was low.*

*Why do you think the diff. stress was low? You find subgrains that indicate stresses as high as 4MPa or as low as 0.4Mpa in the same thin-section? Do you think that it was low, because it shows primarily features indicative for pressure solution-precipiation (PS) creep?*

**Response:** We corrected: "The darker layers are inferred to contain more impurities and have higher viscosities. However, to better quantify the amount of impurities more samples and analysis are needed. We consider the dark salt to be more competent than the white salt based on field and fold shape observations. We note that the dark salt itself consisting of claystone particles and halite grains can have internal competency contrasts (Fig. 11e), which are not resolved in our numerical models. We note that pressure solution microstructure does not provide information on deviatoric stress (solution-precipitation creep does not form subgrains), so we have no way to test if the layers inferred to have had higher viscosities actually deformed under higher deviatoric stress. We only note that deviatoric stress was very low (less than 1 MPa) because of the dominant solution-precipitation creep deformation mechanism."

*Q80: Lines 581: Change "on various of scales" to "on a variety of scales"*

**Response:** We modified it as "at various scales".

*Q81: Lines 639-643: delete*

**Response:** We deleted only the first sentence. We believe that the information about the limitation of our project, i.e., unknown detailed relation between the impurity content and the shades of grey of the layers, is important and should not be removed from the manuscript.

*Q82: Lines 666: anisotropic in terms of viscosity?*

*"anisotropy" expresses uneven properties in different directions,*

*you need to link "anisotropy" with a physical property*

*E.g. anisotropy of seismic velocity, anisotropy of magnetic susceptibility, anisotropy of tensional strength etc...*

**Response:** By anisotropy, we mean mechanical anisotropy. We added a new subsection in the manuscript (Section 7.6) that discusses this issue in more detail.

---

## Author Response (AR2)

Dear Editor,

We would like to thank you for this comments. We believe that the thermal effects are not important factors that could change the rheology of salt and also influence the fold shapes in the Ocnele Mari salt mine. Thus, we modified the Section 4 and 7.2 and referred the work of Hobbs et al. (2008).

On behalf of all authors,
Marta Adamuszek